# Serial Lift-Out: sampling the molecular anatomy of whole organisms

Oda Helene Schiøtz ●[1,6], Christoph J. O. Kaiser ●[1,6], Sven Klumpe ●[1,6]✉, Dustin R. Morado[2,5], Matthias Poege[3], Jonathan Schneider[3], Florian Beck[1], David P. Klebl[2], Christopher Thompson[4] & Jürgen M. Plitzko ●[1]✉

Cryo-focused ion beam milling of frozen-hydrated cells and subsequent cryo-electron tomography (cryo-ET) has enabled the structural elucidation of macromolecular complexes directly inside cells. Application of the technique to multicellular organisms and tissues, however, is still limited by sample preparation. While high-pressure freezing enables the vitrification of thicker samples, it prolongs subsequent preparation due to increased thinning times and the need for extraction procedures. Additionally, thinning removes large portions of the specimen, restricting the imageable volume to the thickness of the final lamella, typically <300 nm. Here we introduce Serial Lift-Out, an enhanced lift-out technique that increases throughput and obtainable contextual information by preparing multiple sections from single transfers. We apply Serial Lift-Out to *Caenorhabditis elegans* L1 larvae, yielding a cryo-ET dataset sampling the worm's anterior–posterior axis, and resolve its ribosome structure to 7 Å and a subregion of the 11-protofilament microtubule to 13 Å, illustrating how Serial Lift-Out enables the study of multicellular molecular anatomy.

Single-particle analysis by cryo-transmission electron microscopy (cryo-TEM) has become a key technique to study the structure of isolated biological macromolecules at high resolution[1]. While single-particle analysis routinely reaches resolutions at which protein side chains can be fitted unambiguously, the reductionist approach of studying protein complexes in vitro loses all information concerning their 'molecular sociology': the interaction of molecular complexes in their natural environment[2]. Conversely, in situ cryo-ET allows for the reconstruction of pleomorphic structures such as the crowded environment of the cell at molecular resolution, maintaining the interaction and localization of protein complexes within the biological system[3–8].

One of the primary factors limiting the resolution of cryo-TEM is inelastic scattering. As the mean free path of an electron in vitrified biological samples is about 300–400 nm, samples beyond the size of viruses and small prokaryotic cells are generally too thick for cryo-ET[7,9]. As a consequence, two main sample-thinning methods to obtain electron-transparent specimens for cryo-ET have been developed: cryo-ultramicrotomy and cryo-focused ion beam (FIB) milling. Cryo-ultramicrotomy encompasses the thin sectioning of vitreous cells and tissues with a diamond knife[10]. The shearing forces at the knife's edge, however, cause mechanical artifacts such as crevices and compression in the resulting sections[11]. More recently, the FIB instrument has been widely adopted for sample thinning at cryogenic temperatures. While not completely free of damage[12,13], the technique bypasses the mechanical artifacts of cryo-ultramicrotomy[14,15] and has been shown to yield data that can allow the elucidation of ribosomes to side-chain resolution[8]. The automation of lamella preparation by cryo-FIB milling has also reduced the need for user expertise and manual intervention[16–18].

[1]Research Group CryoEM Technology, Max Planck Institute of Biochemistry, Martinsried, Germany. [2]Department of Cell and Virus Structure, Max Planck Institute of Biochemistry, Martinsried, Germany. [3]Department of Molecular Structural Biology, Max Planck Institute of Biochemistry, Martinsried, Germany. [4]Materials and Structural Analysis, Thermo Fisher Scientific, Eindhoven, the Netherlands. [5]Present address: Department for Biochemistry and Biophysics, Science for Life Laboratory, Stockholm University, Stockholm, Sweden. [6]These authors contributed equally: Oda Helene Schiøtz, Christoph J. O. Kaiser, Sven Klumpe. ✉e-mail: klumpe@biochem.mpg.de; plitzko@biochem.mpg.de

Before thinning, the sample must be cryogenically fixed by cooling at a sufficiently high rate to prevent ice crystal formation, resulting in a vitrified sample. There are two main methods available for vitrification: plunge freezing, in which the sample is immersed at ambient pressure into liquid ethane or an ethane–propane mixture[19], and high-pressure freezing (HPF), in which the sample is cooled with a jet of liquid nitrogen at a pressure of ~2,000 bar. While the former yields samples that are easily milled with a FIB, the sample thickness that can reliably be vitrified is limited and only the smallest cells vitrify entirely. HPF, on the other hand, allows for the vitrification of samples up to a thickness of roughly 200 μm[20].

Consequently, HPF greatly expands the size range of biological samples that can be vitrified but comes at the cost of embedding the specimen in a thick layer of ice defined by the depth of the freezing receptacle. This increased sample thickness leads to longer milling times. While samples up to a thickness of about 50 μm can be prepared by milling lamellae directly on the grid following the 'waffle' method[21] or alternative freezing approaches using 2-methylpentane[22,23], lamellae from thicker samples have, to date, only been prepared by cryo-lift-out[18,24–28].

Cryo-lift-out refers to the extraction of the material for lamella preparation from a bulk HPF sample and subsequent transfer and attachment to a lift-out receiver grid, conventionally a half-grid[29]. Two main types of micromanipulator devices are currently available for cryo-lift-out: sharp needles and a cryo-gripper[30]. Initially, trenches are milled around the area of interest, leaving it connected to the bulk material on a single side. For specimens in HPF sample carriers, the material must additionally be cleared from below. After these preparatory steps, the lift-out device is attached to the volume to be extracted. The remaining connection to the bulk material is removed, the micromanipulator is used to transfer the extracted volume, and redeposition milling is used to attach the volume to the receiver grid[31]. Finally, an electron-transparent lamella is prepared[32].

While widely used in materials science at room temperature[29], cryo-lift-out of biological samples has remained primarily proof of concept[18,24,26,28]. This is due to a number of factors, for example, the need for cooled micromanipulator devices and required accompanying workflow adaptations, resulting in limited throughput and problems with lamella loss during transfer to the transmission electron microscope[26].

Another limitation to cryo-lift-out as well as on-grid lamella preparation is the loss of contextual information. Only a fraction of the sample volume (~1% for larger eukaryotic cells, <1% for multicellular specimens) ends up inside the final lamella for cryo-ET data acquisition. Techniques exist that are capable of capturing larger volumetric data and tracking morphology at comparatively large volume scales. Examples of such techniques are X-ray tomography and various volume electron microscopy techniques: serial FIB milling and scanning electron microscopy (SEM) or serial sectioning of plastic-embedded samples imaged by SEM, TEM or scanning TEM[33,34]. These techniques, however, currently cannot achieve the resolution attainable by cryo-ET at an equivalent sample-preservation state due to physical limitations in imaging or the necessity of fixation and contrasting.

In this work, we describe a new cryo-lift-out approach that creates a series of lamellae from one lift-out volume, a method we term Serial Lift-Out. Inspired by diamond knife serial sectioning, Serial Lift-Out retains more contextual information than previous procedures and increases the throughput of cryo-lift-out by an order of magnitude. It sets the stage for the study of multicellular organisms and tissues by cryo-ET, applications that previously seemed practically impossible[28]. We demonstrate Serial Lift-Out on high-pressure-frozen *C. elegans* L1 stage larvae, sampling their ultrastructure along the anteroposterior axis by cryo-ET. From the resulting dataset, we reconstruct the nematode's ribosome to a resolution of 7 Å and a region of the 11-protofilament microtubule to 13 Å by subtomogram averaging,

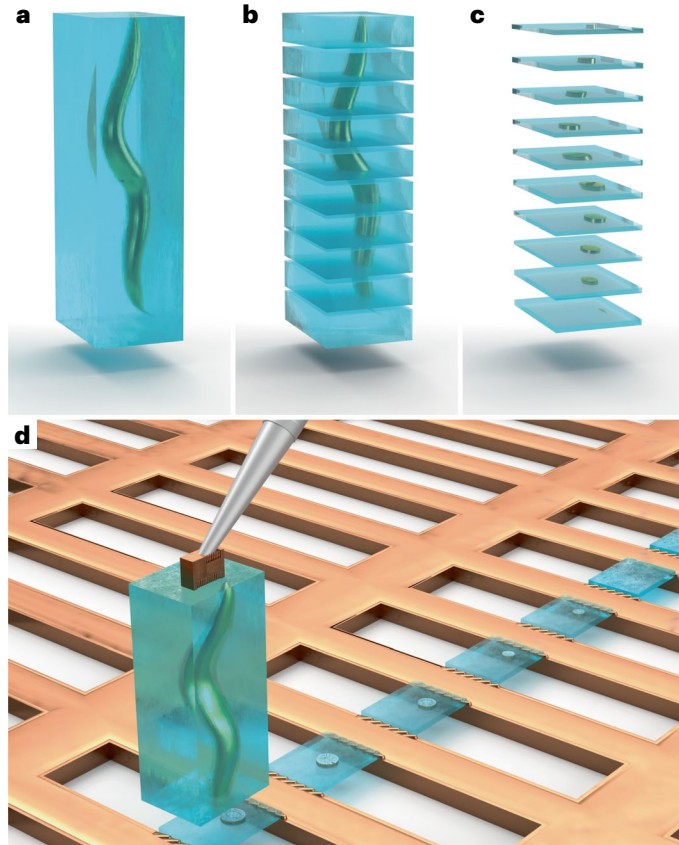

**Fig. 1 | A schematic of Serial Lift-Out.** An illustration of the Serial Lift-Out method, exemplified by the process being performed on a *C. elegans* L1 larva embedded in an extracted volume of vitreous ice (blue). **a**, The extracted volume containing the larva. **b**,**c**, Schematic representation of the resulting Serial Lift-Out sections (**b**) and lamellae (**c**), respectively. **d**, Illustration of the double-sided attachment Serial Lift-Out procedure. The extracted volume shown in **a** is attached to the lift-out needle via a copper block adaptor and transferred to a rectangular mesh receiver grid. Several sections are shown, obtained by the repetition of the attachment of the bottom part of the volume to the grid bars and subsequent sectioning.

exemplifying the enormous potential of Serial Lift-Out for the study of the molecular anatomy of multicellular systems.

## Results

### The concept behind Serial Lift-Out

The most time-consuming steps in cryo-lift-out are the preparation of the extraction sites and the transfer of the extracted volume to the receiving grid. Each repetition of the lift-out cycle therefore adds substantial time investment with low yield. Scaling up the extracted volume and producing multiple lamellae from a single lift-out bypasses this repetitive, time-consuming trench milling and transfer, increases throughput and, foremost, provides more contextual information about the targeted material.

A specific implementation of this concept is the extraction of an entire L1 larva of *C. elegans*, followed by repeated steps of attachment to the receiving grid, sectioning and transfer of the remaining volume to the next attachment site (Fig. 1). While material is still lost during sectioning and thinning, a much larger fraction of the worm is made accessible to cryo-TEM data acquisition.

Previous lift-out approaches extracted volumes approximately the size of the final lamella (Extended Data Fig. 1a), with some excess material for stability during transfer and attachment. To section multiple lamellae from a single cryo-lift-out transfer, the extracted volume needed to be increased. To ease the manipulation of such large

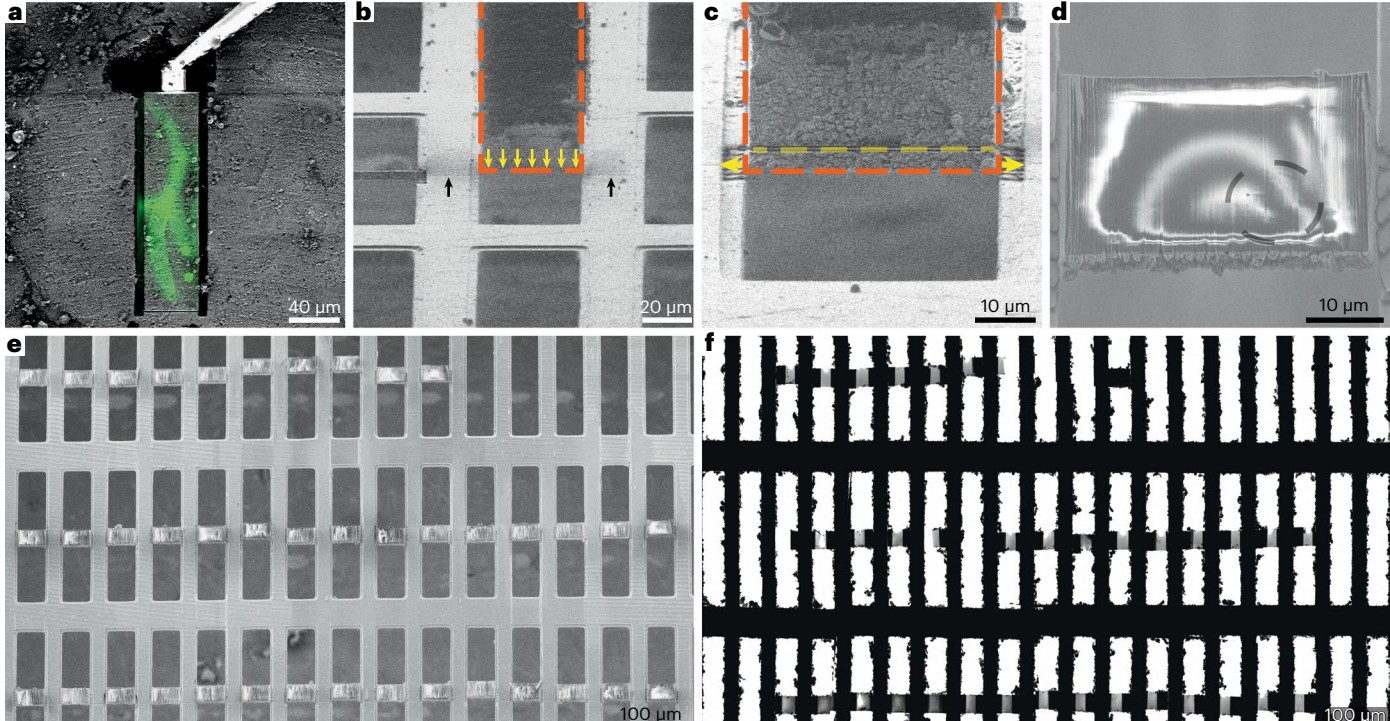

**Fig. 2 | A workflow for double-sided attachment Serial Lift-Out. a**, FIB image of the prepared extraction site with overlaid correlated fluorescence data (green) indicating the larva being targeted. The micromanipulator is attached to the extraction volume by redeposition from the copper adaptor (trench-milling orientation). **b**, The extracted volume (orange dashed line) is lowered into position between two grid bars in lamella-milling orientation. The lower front edge of the volume (yellow arrows) is aligned to the pre-milled line mark (black arrows). **c**, Double-sided attachment by redeposition from the grid bars (yellow arrows indicate direction of milling), followed by line pattern milling, releasing the section of a desired thickness (dashed yellow line). The orange dashed line indicates the outline of the extracted volume. **d**, SEM image of a typical section after being released from the extracted volume. The black dashed line indicates the outline of the worm cross-section. **e,f**, SEM (**e**) and TEM (**f**) overview images of the 40 double-sided attached sections and the resulting lamellae. Supplementary Video 1 summarizes the process. The data presented in this figure stem from experiment 2.

volumes, we introduced a copper block that acts as an adaptor between the needle and the extracted volume. This copper block is created from the receiver grid and attached to the needle before lift-out (Extended Data Figs. 2 and 3). As copper has higher redeposition rates than the tungsten of the needle, the copper block adaptor results in substantially more resilient 'welding' of the specimen to the micromanipulator.

Another volume-limiting factor is the ablation rate of the ion beam. Most cryo-FIB machines are equipped with a gallium ion source, rendering it impractical to mill beyond 50 µm in depth. In addition, for samples frozen in HPF carriers, preparation requires an undercut, the removal of material below the extracted volume, to detach the extraction volume from the bulk. In combination, these factors generally result in maximum extraction volumes of approximately 20 µm × 20 µm × 10–30 µm (length × width × height; Extended Data Fig. 1b).

To extract larger volumes, we performed lift-out on an HPF 'waffle'-type sample. HPF 'waffle' samples are prepared by freezing the sample on a grid that is sandwiched between HPF carriers[21]. The final thickness is therefore defined by the type of grid (and spacer) being used during freezing. For a 25-µm-thick specimen, extraction volumes of up to 200 µm × 40 µm × 25 µm can easily be obtained by performing lift-out with the sample surface oriented perpendicular to the ion beam (Extended Data Fig. 1c). The same orientation is used during trench milling and will be referred to as the 'trench-milling orientation' (Extended Data Fig. 3c,f). Such large volumes extracted at the trench-milling orientation can yield many sections from a single lift-out, which can be subsequently thinned to lamellae for cryo-ET data acquisition (Fig. 1b,c).

### Application of Serial Lift-Out to *C. elegans* L1 larvae

To assess the feasibility of obtaining serial lamellae from a single lift-out transfer, we performed the procedure on *C. elegans* L1 larvae that had been vitrified using a modified protocol of the 'waffle' method (Fig. 2). The sample contained many L1 larvae embedded in an approximately 25-µm-thick layer of ice (Supplementary Fig. 1). Three sites from three different grids were selected for preparation by correlating cryo-fluorescence light microscopy with SEM and FIB view images. Following the geometry described in Extended Data Fig. 1c, the sites were prepared by milling trenches around the larva, yielding extraction volumes of 110 µm × 30 µm × 25 µm and 180 µm × 40 µm × 25 µm (Fig. 2a and Extended Data Fig. 4a). To extract the volumes, the needle with the copper adaptor (Fig. 2a and Extended Data Fig. 4b) was attached to the sample using redeposition: copper material redepositing onto the surface of the extraction volume during milling. The extraction volume was subsequently milled free from the bulk material and lifted out. The volume was then transferred to the receiver grid. Two different attachment strategies were explored, with the resulting lamellae either being attached on one side (single-sided attachment, experiment 1) or two sides (double-sided attachment, experiments 2 and 3).

For single-sided attachment, a modified grid based on customizing a standard 100-mesh copper grid (Supplementary Fig. 2) was used as a receiver grid. Removing every second row of grid bars yielded an array of trimmed grid bars that resemble the pins of a standard lift-out half-grid (Supplementary Fig. 2e). Alternatively, a copper grid with rectangular meshes can be used for double-sided attachment of volumes roughly as wide as the mesh (~40 µm for the 400/100 rectangular

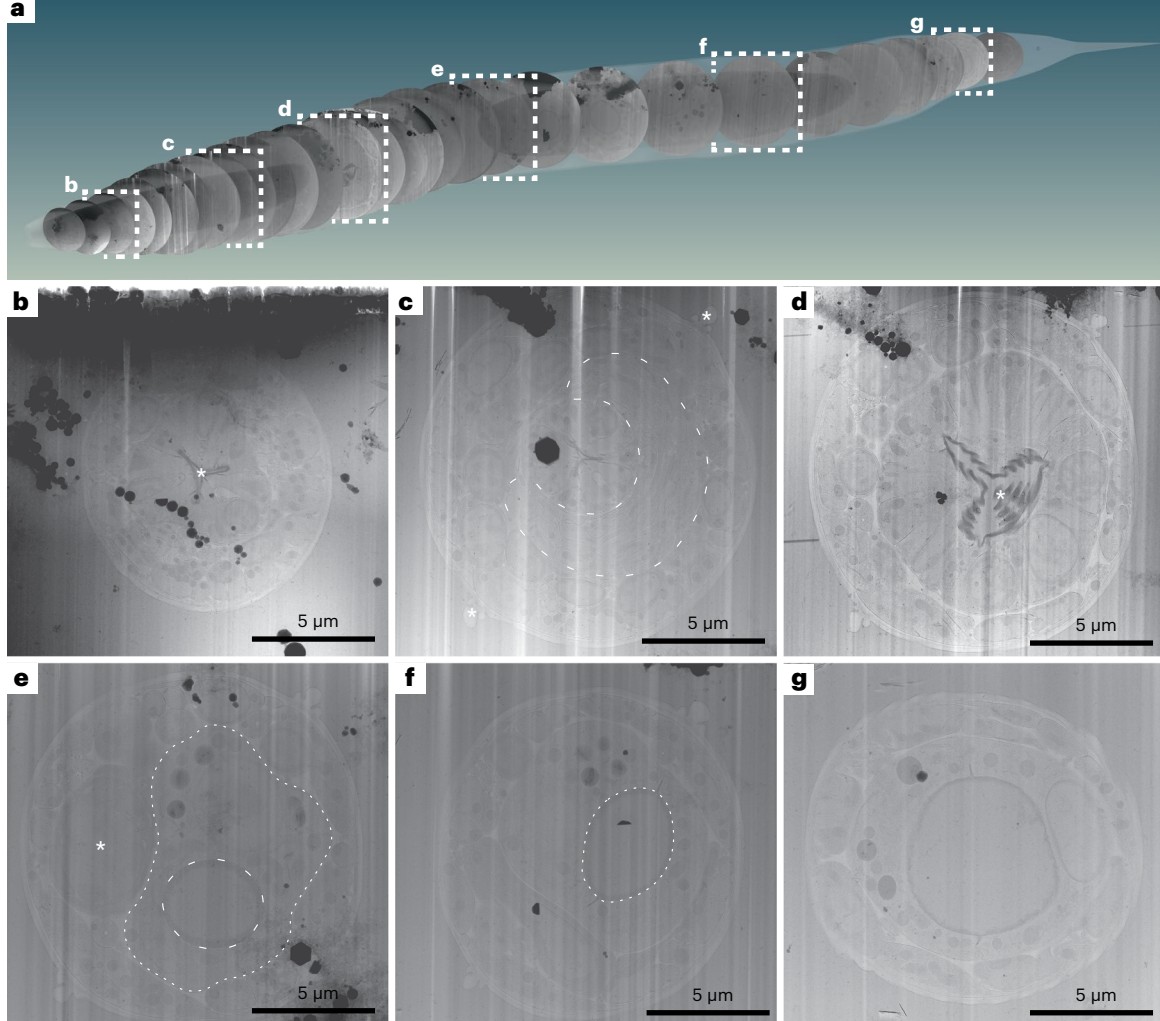

**Fig. 3 | Lamella TEM overviews sample the anatomy of a *C. elegans* L1 larva along the anterior–posterior body axis.** Native tissue scattering contrast is sufficient to extract a considerable amount of anatomical information from low-magnification TEM overviews of the lamellae generated in the double-sided-attachment Serial Lift-Out experiment. **a**, Schematic representation of 29 body transverse-section overviews obtained from the final lamellae along the anteroposterior body axis of an L1 larva. Anterior is located to the left; posterior is to the right. The OpenWorm project model of the adult *C. elegans* worm was used for illustrating the possibility of back mapping, as a cellular model of the L1 larva only exists for its head and not the entire body[39]. The cross-sections were cropped from lamella-overview images and mapped back to a position derived from the known sectioning distance and anatomical features discernible in the corresponding cross-section. Dashed white frames indicate overviews with corresponding magnified representations in **b**–**g**. **b**, This lamella originates from ~15 μm along the anterior–posterior axis. Clearly visible are the three lobes of the anterior pharyngeal lumen in the center of the worm cross-section (asterisk) and the relatively electron-dense pharyngeal lining. **c**, Overview from the anterior part of the pharyngeal isthmus ~42 μm along the anteroposterior axis. Note the nerve ring (dashed line) surrounding the central pharynx. Additionally, the alae (asterisks) running along the left and right lateral sides of the worm become obvious. **d**, Overview of a lamella of roughly the center of the posterior pharyngeal bulb region. The central grinder organ is clearly discernible (asterisk). This section is positioned ~65 μm along the anterior–posterior axis. **e**, A section roughly mid-body. The intestinal lumen (dashed line) and intestinal cells (dotted line) are obvious. The darker cell slightly left of the body center is likely one of the gonadal primordial cells (asterisk). The section is from ~115 μm along the anteroposterior axis. **f**, In this mid-body section, the intestinal lumen (dashed line) can again be clearly discerned. The section can be mapped to ~132 μm along the anteroposterior axis. **g**, Section showing the intestinal lumen at ~155 μm along the anteroposterior axis. The data presented in this figure stem from experiment 2.

mesh grids used here; Fig. 2b). Double-sided attachment has the added benefit of increased section stability by avoiding the free-standing side of the lamella (Extended Data Fig. 4f).

The volume's lower front edge was precisely aligned to the front edge of the attachment pin (Extended Data Fig. 4c) or a previously prepared alignment line pattern milled onto the receiver grid (Fig. 2b). Next, redeposition milling was used to attach the lower part of the volume to the grid bar(s) (Fig. 2c and Extended Data Fig. 4d, yellow arrows). After attachment, the lower part of the target material was separated from the extracted volume using a line pattern (Fig. 2c and Extended Data Fig. 4d, yellow dashed line), leaving a ~4-μm-thick section

(Fig. 2d and Extended Data Fig. 4e,f). Once sectioned, the remaining volume attached to the lift-out needle was transferred to the next attachment site. This procedure was iterated until no material remained. As a result, many sections were produced from a single cryo-lift-out transfer (Fig. 2e,f and Extended Data Fig. 4g).

To assess the general applicability of the approach to non-'waffle'-type samples, an additional experiment was performed on *Drosophila melanogaster* egg chambers, high-pressure frozen in standard sample carriers. From a ~20-μm-long and ~15-μm-deep extracted volume, we produced sections ~1–2 μm in thickness (Extended Data Fig. 5).

### Sampling the *C. elegans* L1 larva at molecular resolution

We prepared dozens of sections along the anteroposterior axis of *C. elegans* L1 larvae: 12 single-sided-attached (Extended Data Fig. 4g) and 40 double-sided-attached (Fig. 2e and Supplementary Video 2) lamellae. After transfer to the transmission electron microscope, eight of 12 and 32 of 40 lamellae were recovered, respectively (Supplementary Fig. 3a and Extended Data Fig. 6a). Lamella loss during transfer is common due to manual grid-handling steps. The increased rate of successfully transferred lamellae from 66% for single-sided attachment to 80% for double-sided attachment is indicative of the increased lamella stability of double-sided attachment.

From these successfully transferred lamellae, overview maps were recorded (approximately 20 µm × 25 µm in size; Supplementary Fig. 3 and Extended Data Fig. 6), showing larval cross-sections. These overviews allowed the identification and assignment of anatomical structures and tissues such as the pharynx, body wall muscle cells, neurons, the hypodermis and seam cells. Larger organelles such as nuclei, mitochondria, Golgi cisternae, storage granules, junctional regions, bundles of actin filaments and microtubules were clearly discernible (Fig. 3b–g and Supplementary Video 3). Given the stereotypic body plan of *C. elegans*, the information obtained from the overviews together with the sectioning thickness was used to determine the approximate location of the sections in the worm, as schematically shown in Fig. 3a.

The biological area within each lamella can, due to the circular nature of cross-sections, be estimated by $\pi r^2$, where the average radius is around 5 µm, resulting in a mean of 78.5 µm$^2$ in imageable area per lamella. For the single-sided-attachment experiment, the total imageable area was ~630 µm$^2$ and was subjected to the collection of 56 tilt series, each with a field of view ~1.2 µm × 1.2 µm in size. The double-sided-attachment experiment yielded ~2,500 µm$^2$ in imageable area, from which a total of 1,012 tilt series with a field of view ~750 nm × 750 nm in size were collected. The double-sided attachment experiment allowed us to sample a larger fraction of tissues and cell types along the anteroposterior body axis of the *C. elegans* L1 larva.

To illustrate the information level already present at intermediate magnification (11,500×), we manually segmented a representative cross-section (Fig. 4a). The segmentation illustrates that the cuticle clearly delineates the body cross-section. Body wall muscle cells are obvious due to their pronounced distal actomyosin pattern. The pharyngeal lumen's trilobal structure demarcates the body center and is surrounded by the three alternating pharyngeal muscle and marginal cells. Pharyngeal neurons and gland cell process cross-sections are embedded in the pharyngeal muscle cells. Nuclei are easy to discern due to their ribosome-decorated double membrane and their denser, less granular interior. Neurons in general can be discerned by their appearance as round or tubular cells, often grouped in bundles. These clearly interpretable overviews allow targeting of tissue-specific cellular structures or cellular protein complexes with known location (for example, sarcomeric proteins).

### Quality assessment and subtomogram analysis

To assess the quality of lamella thinning, tomogram thickness was measured for all 56 tomograms from the single-sided-attachment experiment and a random subset of 132 tomograms for the double-sided-attachment experiment. The thickness for single-sided attachment was 255 ± 126 nm (experiment 1). The thickness of the tomograms from the double-sided-attachment experiment was more uniform, 303 ± 40 nm (experiment 2) and 252 ± 36 nm (experiment 3) (Extended Data Fig. 7). The broader thickness distribution in the single-sided-attachment experiment most likely stems from lamella bending and movement during milling due to the free-standing edge of the lamella.

The tilt series we obtained allowed us to reconstruct the three-dimensional (3D) cellular architecture of cell types in different tissues such as the neuronal nuclear periphery (Fig. 4b), pharyngeal marginal cells and neuronal cell bodies (Fig. 4c), body wall muscle, the hypodermis and the collagenous network of the cuticle (Fig. 4d).

The sectioning plane obtained when performing Serial Lift-Out with 'waffle'-type samples provides worm cross-sections. For a number of structures, such as nuclei, the difference between oblique longitudinal and transverse sectioning is minimal (Fig. 4a and Extended Data Fig. 8). When investigating anisotropic structures such as the body wall muscle or the pharynx, however, sectioning direction can greatly impact the interpretation of higher-order structure. In longitudinal sections, sarcomeres, located in body wall muscle, appear as filaments running along the image plane. These filaments show an orientational distribution strongly biased toward the side view. By contrast, tomograms from Serial Lift-Out show transverse sections of the body wall muscle. In this sectioning plane, actin and myosin run often perpendicular to the image plane and can be discerned as small- and medium-sized puncta. This more clearly reveals the organization of actomyosin bundling and holds the potential for further analysis of their packing (Fig. 4d).

To assess the quality of the data acquired, we picked 35,350 80S ribosome particles from 200 tomograms from the double-sided-attachment experiment. Subtomogram averaging and classification yielded a structure at a resolution of 6.9 Å (gold-standard Fourier shell correlation, GSFSC; Fig. 5a and Extended Data Fig. 9a). The resolution was likely limited by lamella thickness, supported by the fact that the contrast transfer function (CTF) could not be fit beyond 6 Å.

Classification yielded four different subpopulations in various translational states. When comparing these states to the recently published ribosome state landscape from *Dictyostelium discoideum*[8], resemblances are apparent to the initiation state with an occupied P site (Fig. 5b, class 2), states with occupied A and P sites (Fig. 5b, class 3) and elongation factor-bound states (Fig. 5b, classes 1 and 4 and Extended Data Fig. 9b).

Furthermore, we found cell type-specific differences in microtubule structure (Fig. 5c). Microtubules with distinct protofilament numbers of 11 and 15 were identified (Fig. 5c–e), the latter originating from a single cell likely to be a touch-receptor cell. To assess whether these filaments were amenable to subtomogram averaging, we traced microtubule filaments and focused on the 11-protofilament microtubule structure. From 28 filaments, we obtained an initial average of a larger fraction of the microtubule (Extended Data Fig. 9c,d). To account for flexibility within the microtubule structure, we oversampled the filaments to focus the alignment on the tubulin subunits (Extended Data Fig. 9e). After refinement, we obtained a 13 Å structure of the focused microtubule subregion (Fig. 5f and Extended Data Fig. 9f).

## Discussion

In the past, cryo-lift-out has been hampered by limited throughput and a low overall success rate. Therefore, the technique has even been deemed 'practically impossible' or at least 'merely difficult' in the literature[28]. The fact that the existing literature has not advanced beyond proof-of-principle experiments underlines this evaluation. The study of multicellular organisms and tissues by cryo-ET, however, holds enormous potential for biological discovery, and technical advances are thus needed. We anticipate the combination of recent hardware and workflow improvements[30–32] with the increased throughput of Serial Lift-Out to make cryo-ET data acquisition from high-pressure-frozen material more attainable. In addition, plasma FIB technology, while available for some years, was recently introduced to the cryo-FIB community[13,35,36] and may further improve cryo-lift-out throughput by increased ablation rates.

Cryo-FIB lamella-milling protocols developed to date remove most of the cell during lamella preparation. The final lamella represents only

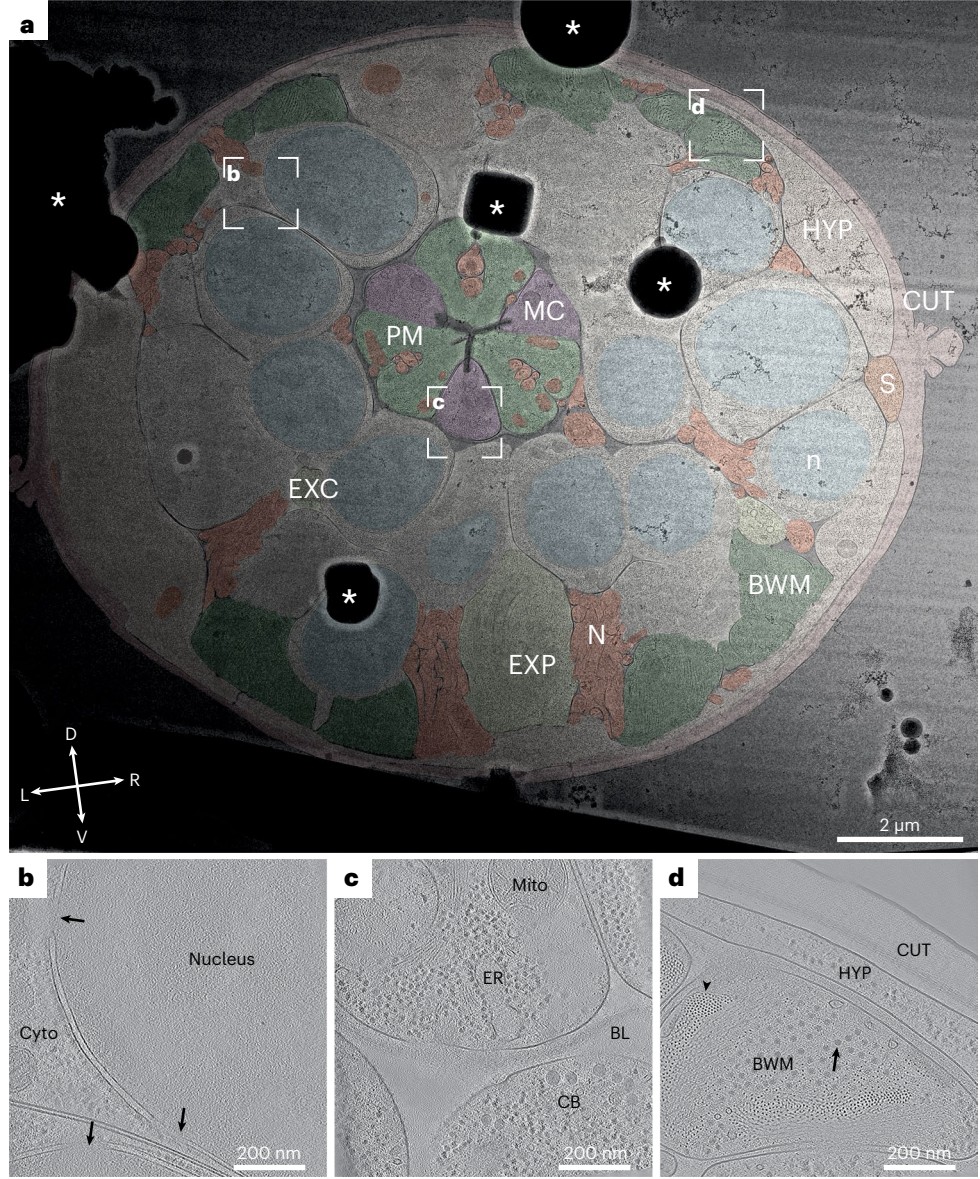

**Fig. 4 | Representative overview and tomograms of the L1 larval pharyngeal isthmus region. a**, Overview montage (magnification, ×11,500) of a head region lamella. This section is located within the pharyngeal isthmus just posterior to the nerve ring. Cell types are colored according to the WormAtlas color code[46]; for nuclei and mitochondria, arbitrary colors were chosen (HYP, hypodermis; S, seam cells; PM, pharyngeal muscle; MC, marginal cells; EXP, excretory pore; EXC, excretory canal; N, neuronal tissue; CUT, cuticle; n, nucleus; BWM, body wall muscle. White dashed rectangles show the positions of the tilt series acquired, and corresponding reconstructions are shown in **b**–**d**. Asterisks indicate ice contamination. The cross indicates dorsal (D)–ventral (V) and left (L)–right (R) body axes. Note that the gradient in contrast originates from thickness variations in the lamella with the incident ion beam during milling coming from the right. **b**, Tomographic slice of the perinuclear region of most likely a neuronal cell. The nucleus exhibits a different granularity and density than the cytoplasm (Cyto),

from which it is separated by the nuclear envelope, which in turn is heavily decorated with ribosomes and contains nuclear pores (arrows). **c**, Tomographic slice of the pharynx. Central in the upper half is a marginal cell containing endoplasmic reticulum (ER) and a mitochondrion (Mito). A neighboring neuronal cell body (CB) is separated from the pharynx by a diffuse density, which is the pharyngeal basal lamina (BL). **d**, Tomographic slice of a body wall muscle cross-section (BWM). Clearly visible are actin filaments in the top view (arrowhead) surrounding a bundle of thick filaments (arrow). The thick filaments are partially interspersed with actin filaments. The muscle cell neighbors a hypodermal cell (HYP), from which it is separated by a space filled with a diffuse density, likely the body wall muscle basal lamina. The next and last layer of the larval body wall is the cuticle. Most notably within this chitinous–collagenous structure, a fence-like array of denser structures can be discerned, likely cuticular collagen. The data presented in this figure stem from experiment 1.

<1% of a eukaryotic cell and even less for multicellular organisms. While a high number of lamellae milled at different heights could restore that lost information through an ensemble average[37], Serial Lift-Out lamellae originate from a single specimen, yielding a more thorough characterization of its morphology. A variety of volume microscopy methods, for example, volume electron microscopy or X-ray tomography, retain this information. They remain, however, limited in resolution when compared to cryo-ET. Serial Lift-Out increases the contextual

information retained in cryo-ET while allowing data acquisition at molecular resolution.

Here, we have shown the ability to section in increments of 1–4 μm (Extended Data Fig. 10). This process has two steps that contribute to specimen loss: sectioning (~300–500 nm) and lamella milling. The latter can be minimized by finer sectioning. Preliminary experiments suggest that thinner sectioning down to 500 nm may be achievable. Reducing the section thickness even further to prepare lamellae

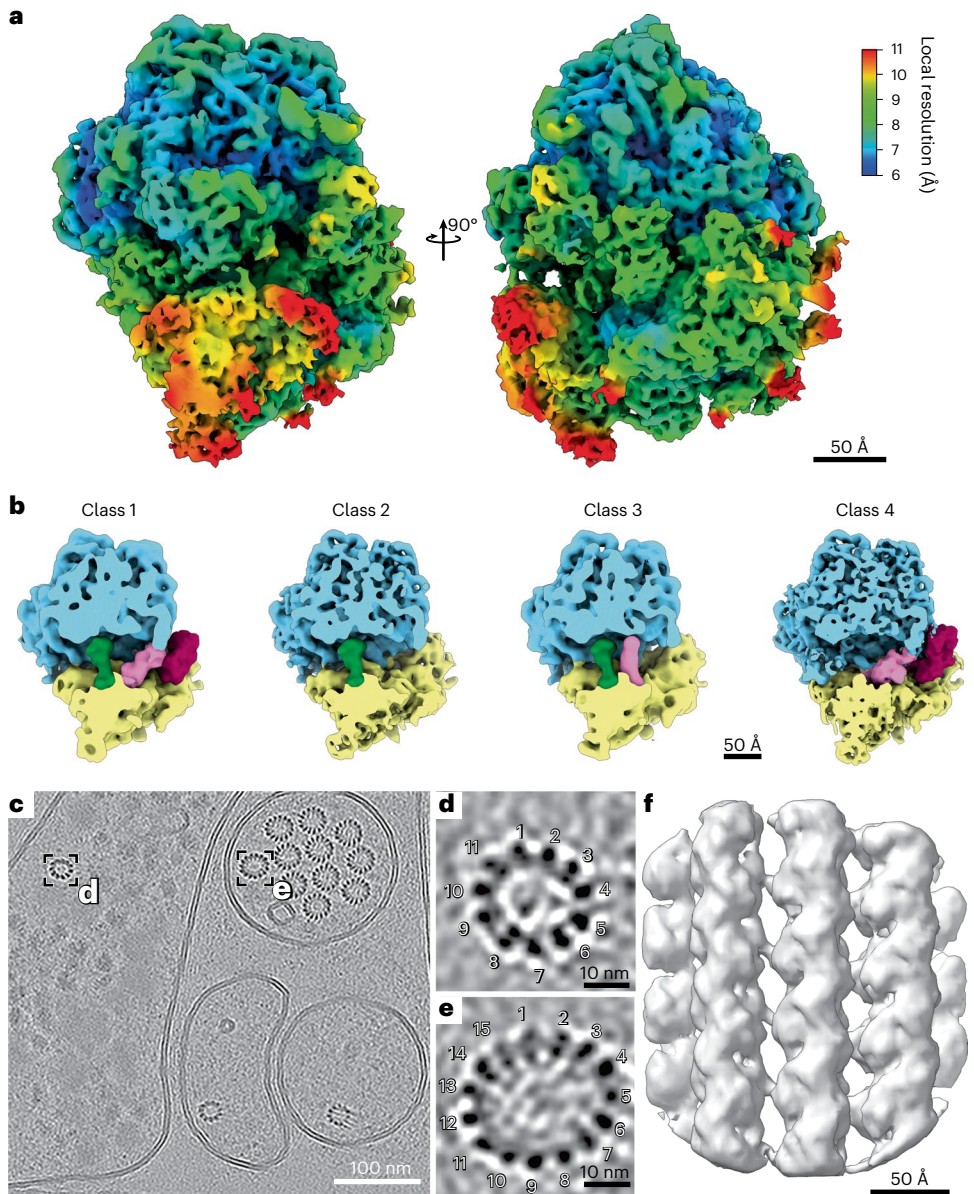

**Fig. 5 | Subtomogram average reconstruction of the *C. elegans* 80S ribosome and the 11-protofilament microtubule from in situ data. a**, The *C. elegans* ribosome reconstructed by subtomogram averaging to a resolution of 6.9 Å (GSFSC). The density map is colored by local resolution. Note that protein α-helices and ribosomal RNA helices are clearly visible at this resolution. **b**, Four different ribosomal states were obtained through subtomogram classification: ribosome class 1 with occupied A, P and EF sites, ribosome class 2 with an occupied P site, ribosome class 3 with occupied A and P sites, ribosome class 4 with occupied A and EF sites. **c**, Tomographic slice of a tomogram collected from, what is likely, the ventral sublateral nerve cord with a touch-receptor neuron (PLML/R). Boxes highlight two different *C. elegans* microtubule assemblies made up of 11 and 15 protofilaments. **d**, Magnified tomogram section of the 11-protofilament microtubule from a hypodermal cell highlighted in **c**. **e**, Magnified tomogram section of the 15-protofilament microtubule from the touch-receptor cell highlighted in **c**. **f**, Subtomogram average of a focused region of the 11-protofilament microtubules resolved to 13 Å. Data presented in **a** and **b** stem from experiment 2; data presented in **c**–**f** stem from experiment 3.

directly from the extracted volume, however, will likely require major technological advances of the FIB instrumentation. Developments in ion beam shaping could become valuable for such endeavors, reducing the material lost during sectioning. Due to the physical basis of ablation in FIB preparation, however, material will always be lost to the milling process itself. Therefore, truly consecutive lamellae, similar to serial sections of plastic-embedded samples, seem unlikely to be attainable.

Nevertheless, the creation of multiple sections within biological material increases the preparation throughput for HPF samples and adds contextual information. For model organisms with a stereotypic and well-described body plan such as *C. elegans* or *Platynereis dumerilii*[38],

Serial Lift-Out sections and tomograms may be mapped back into context using other sources of volumetric data as illustrated in Fig. 3a[33,39]. As a result, back-mapping analysis may enable label-free targeting of features and events that are tissue and cell type specific.

Serial Lift-Out also addresses the challenge that arises when dealing with anisotropic cells and tissues. As the sectioning plane of the specimen can be adapted, fluorescence or FIB–SEM data can inform the preparation of the lift-out site and, in turn, the sectioning angle. This adaptive preparation strategy can give new insights into the molecular architecture, as illustrated by tomograms from transverse sections of muscle cells. When compared to previously obtained longitudinal sections[40–42], the transverse section shows actin–myosin packing from

a new angle, revealing how actin filaments surround myosin filaments. The combination of acquiring cryo-ET data on both transverse and longitudinal sections also holds the potential to improve subtomogram analysis when facing structures with preferential orientation.

In addition to guiding site preparation for lift-out, cryo-correlative light and electron microscopy are more generally used to target specific subcellular events[18,26,43,44]. This technique, however, remains challenging for routine use within larger tissues. Improvements in the operation of integrated light microscopes are therefore necessary to streamline subcellular targeting in cryo-lift-out experiments. Serial Lift-Out, combined with in-chamber light microscopy, could increase the success rate of targeting by reducing the sample thickness used in the correlation. This increases the number of sections and, in turn, targeting attempts and enables one to regularly check the fluorescence signal throughout the milling process.

One limitation of cryo-lift-out is the high amount of ice contamination during transfer. As can be seen in the TEM overviews (Extended Data Fig. 6 and Supplementary Fig. 3), large ice crystals tend to obstruct regions of the lamellae and prevent data acquisition. Serial Lift-Out, similar to on-grid preparations, compensates for the loss in imageable area through a higher yield of lamellae in comparison to previous lift-out techniques. Reducing transfer contamination, however, remains highly desirable, and controlled environments, for example, glove boxes[45] or other technological advances such as vacuum transfer will be needed to maximize the imageable area of cryo-FIB lamellae.

Finally, the analysis of lamella-thickness distributions and lamella-survival rate suggests that double-sided attachment in cryo-lift-out is advantageous. This is likely due to the reduction of lamella bending during milling and greater stability during the transfer to the transmission electron microscope.

With the methodological advances of Serial Lift-Out, the existing hurdles of lift-out have been greatly diminished, enabling data quality, throughput and success rate in cryo-lift-out that permit the mapping of large tissue regions and whole organisms at the molecular level. Tomography and subtomogram averaging on lamellae of an L1 larva obtained with our Serial Lift-Out method elucidated its ribosome to a resolution of 6.9 Å and uncovered four different translational states. Furthermore, we identified 11- and 15-protofilament microtubule structures in situ and averaged a subregion of the 11-protofilament structure to 13 Å. In sum, Serial Lift-Out demonstrates enormous potential to discern and study the molecular anatomy of native tissues and small multicellular organisms.

## Online content

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

## Methods

### Sample vitrification

*C. elegans* strains AM140 (allele rmIs132[unc-54p::Q35::YFP]) and NK2476 (allele qy46[ina-1::mKate+loxP]) were cultivated according to standard methods on rich NGM[47]. To obtain a synchronous population of animals, gravid adult worms were washed from five Petri dishes (6 cm) with M9 medium. The worms were pooled into 15-ml centrifuge tubes and centrifuged at 175*g*. Excess supernatant was removed, and the worm suspension was mixed with bleach solution (3.75 ml of 1 M NaOH, 3.0 ml household bleach, 8.25 ml water[48]) and swiftly vortexed. Bleaching was continued with intermittent vortexing and checking on a stereomicroscope. As soon as roughly 50% of worms appeared broken, the bleaching procedure was stopped by the addition of egg buffer (118 mM NaCl, 48 mM KCl, 2 mM CaCl$_2$, 2 mM MgCl$_2$, 25 mM HEPES, pH 7.3) and centrifugation at 175*g* for 1 min. The supernatant was swiftly removed, and the egg–worm carcass solution was washed another four times with egg buffer. The egg–worm carcass mixture was then floated on a 60% sucrose solution, and purified eggs were withdrawn from the surface[49]. The egg solution was washed with M9 buffer, and larvae were allowed to hatch at 20 °C for 24 h.

This population of developmentally arrested L1 larvae was used for vitrification, carried out following a modified version of Mahamid et al.'s method[25] on a Leica EM ICE (Leica Microsystems). Type B sample carriers (Leica Microsystems) with a diameter of 6 mm were coated with a separating layer of cetyl palmitate solution (0.5% (wt/vol) in diethyl ether) by dipping the carriers briefly into the solution and placing them, cavity side down, on filter paper to let the solvent evaporate. A 2-μl drop of 20% (wt/vol) Ficoll 400 in M9 was applied to the flat side of the sample carrier, and a Formvar-coated grid (50 square mesh or 75 hexagonal mesh) was floated on this drop with the support film facing the liquid. Any excess cryoprotectant below the grid was wicked away using pieces of Whatman no. 1 filter paper (Whatman). Next, synchronized L1 larvae were mixed with an equal amount of 40% (wt/vol) Ficoll 400 in M9 medium to reach a final concentration of 20% cryoprotectant. Two microliters of this sample solution were applied onto the grid, and the sandwich was completed by addition of a second 6-mm type B sample carrier, cavity side up. The sample was immediately frozen with high pressure. Grids were removed from the HPF sample carriers and stored in liquid N$_2$ until further use. EM grids were clipped into Thermo Fisher Scientific cartridges for FIB milling. To locate the biological material (L1 larvae) in the ice layer, clipped grids were either previously mapped in a Leica SP8 confocal microscope equipped with a cryostage or directly transferred into an FIB–SEM instrument with an integrated light microscope.

Samples from the *D. melanogaster* strain expressing emGFP–NUP358 (emGFP::nup358 PBac{y[+mDint2] = vas-Cas9}VK00027) were prepared as previously described[18,50]. In brief, fly strains were maintained at 22 °C on standard cornmeal agar. Twenty-four hours before dissection of egg chambers, the flies were transferred into a new vial supplemented with yeast paste. Egg chambers were dissected in Schneider's medium. For HPF, 3-mm HPF carriers were soaked in hexadecene and blotted on filter paper. Depending on the egg chamber stage, the dissected material was subsequently transferred into the 100-μm or 150-μm cavity of a 3-mm type A or type C HPF sample carrier (Engineering Office M. Wohlwend), respectively. The filling medium was 20% Ficoll 70 in Schneider's medium. The flat surface of a type B HPF planchette was used to close the HPF assembly for freezing in the Leica EM ICE high-pressure freezer (Leica Microsystems). The sample was subsequently pre-trimmed using a 45° diamond knife (Diatome) in a cryo-microtome (EM UC6/FC6 cryo-microtome, Leica Microsystems) at a temperature of −170 °C and used for further FIB preparation.

### Cryo-fluorescence microscopy

Grids were clipped in Thermo Fisher Scientific cartridges and imaged using a Leica TCS SP8 laser confocal microscope (Leica Microsystems) fitted with a Leica cryostage[51]. Imaging was performed with a ×50 0.90-NA objective and HyD (fluorescence) or PMT (reflection) detectors using LAS X (3.5.5.19976, Leica Microsystems).

Tile set montages of the NK2476 strain sample were collected using a pinhole size of 4.85 airy units (AU) and a voxel size of 0.578 μm × 0.578 μm × 2.998 μm. The tile sets were merged, and maximum-intensity projections for each channel were calculated using LAS X. Autofluorescence imaging was performed for the double-sided-attachment experiment. The 488-nm laser line at 2.5% total power was used for excitation, and emission was detected from 501 nm to 535 nm. The reflection channel was excited at 552 nm and 2% total laser power and detected with an emission range of 547 nm to 557 nm.

Tile set montages of the AM140 strain were collected at a pinhole size of 1 AU and a voxel size of 0.578 μm × 0.578 μm × 1.03 μm. The tile sets were merged, and maximum-intensity projections for each channel were calculated using LAS X. Q35–YFP was excited at 488 nm and 0.5% total laser power, and emission was detected from 509 nm to 551 nm. Reflection was recorded with 638-nm excitation at 0.83% total laser power and the emission range set to 633 nm to 643 nm.

### Single-sided attachment Serial Lift-Out

**Lift-in grid preparation.** Copper grids (100 square mesh, Agar Scientific) were assembled into Thermo Fisher Scientific cartridges. The cartridges were marked in line with grid bars to aid in orientation alignment during sample loading. The grids were then loaded into an FIB–SEM 45° pre-tilt cryo-shuttle such that the grid bars were aligned vertically and horizontally.

Once loaded, 14-pin grids were prepared by rotating the grid plane to be normal to the ion beam (trench-milling orientation) and milling out two combs of horizontal grid bars. This yielded grids with 14 pins that could be used for 28 lamella attachments (Supplementary Fig. 2). A horizontal grid bar was used to prepare 20-μm × 10-μm × 10-μm copper blocks for attachment to the lift-out needle by redeposition milling using single-pass regular cross-sections directed away from the needle (Extended Data Figs. 2 and 10a). Note that the copper block attached to the needle can be re-used several times. The lower section of the copper block used for the previous attachment can be milled away, leaving a clean surface for the next attachment.

As the FIB–SEM instrument does not have to be cooled during lift-in grid milling, we recommend the preparation of several lift-in grids ahead of time to reduce workload during the Serial Lift-Out session.

**Lift-out, sectioning and milling procedure.** Serial Lift-Out was performed on an Aquilos 2 FIB–SEM instrument (Thermo Fisher Scientific) equipped with an EasyLift system. To facilitate lift-out, the EasyLift needle was modified by attaching a copper block adaptor (Lift-in grid preparation) using redeposition. The copper adaptor has a higher sputter yield and thus increased redeposition compared to tungsten and vitreous ice. Additionally, the block increases the surface area available for attachment to the biological sample and reduces the wear of the needle over time.

After copper block attachment, the volume to be extracted was prepared by milling trenches around the region of interest in trench-milling orientation. The fluorescence data acquired from the L1 'waffle' sample were correlated to the ion beam images using MAPS version 3.14 (Thermo Fisher Scientific). The grid was coated with a protective metal–organic platinum layer using a gas injection system (GIS) heated to 27 °C at a stage working distance of 9.72 mm and a stage tilt of 45° for 90 s. Trenches were milled with the grid perpendicular to the ion beam (trench-milling orientation), after which the EasyLift needle was inserted. To increase the redeposition yield for attachment, the copper block needs to be aligned below the sample surface and flush to the surface exposed by trench milling. Redeposition from the copper block onto the extraction volume was achieved using single-pass patterns

**Table 1 | Milling pattern parameters for Serial Lift-Out**

| Step | Pattern type | Beam current | Multipass number | Approximate time |
|---|---|---|---|---|
| Copper block milling | Regular cross-section | 5–65 nA | 4 | 20 min |
| Copper block attachment | Regular cross-section | 1 nA | 1 | 3 min |
| Trench milling | Regular cross-section | 3 nA | 4 | N/A |
| Extraction volume attachment | Regular cross-section | 1 nA | 1 | 3 min |
| Extraction volume release | Line | 3 nA | N/A | 10 min |
| Pin or grid bar attachment | Regular cross-section | 1 nA | 1 | 4–8 min |
| Sectioning | Line | 0.5–1 nA | N/A | 8–12 min |

The table summarizes the parameters used for the milling steps necessary to perform a Serial Lift-Out experiment. The corresponding pattern geometries are illustrated in Extended Data Fig. 10. All patterns were milled at an FIB acceleration voltage of 30.00 kV. Pattern files for Thermo Fisher Scientific instruments are provided in Supplementary Data 1. These templates and the milling parameters may need adjustment for different projects and for FIB–SEM instruments from other manufacturers, specifically concerning differing scanning strategies deployed by the microscope manufacturer. Parameters not applicable to the line patterns are labeled N/A.

of regular cross-sections instead of the default multipass mode to avoid re-milling previously redeposited material. These patterns were placed at the interface of the two surfaces on the copper adaptor with the milling direction for these patterns directed away from the extraction volume (Extended Data Fig. 10b). The extraction volume was then released by milling the last connection to the bulk sample with a line pattern. The extracted volume of 110 μm × 30 μm × 25 μm was subsequently lifted from the bulk sample.

After extraction of the volume, the shuttle was returned to the lamella-milling orientation (18° in a system with a 45° pre-tilt), and the stage was translated to the receiver grid. The lower edge of the extracted volume and the corner of the pin were aligned in both electron and ion beam images. Attachment was performed by redeposition, using single-pass regular cross-section patterns directed away from the extracted volume (Extended Data Fig. 10d). Once attached, the EasyLift system was maneuvered by a 50-nm step horizontally and vertically away from the pin to create a small amount of strain. Subsequently, the lower 4 μm of the volume was sectioned from the rest of the volume using a line pattern. This process was iterated until the entire extracted volume had been sectioned.

### Double-sided attachment Serial Lift-Out

**Lift-in grid preparation.** For double-sided attachment, 100/400 rectangular mesh copper grids (Gilder) were used as the receiver grid. These were clipped into standard Thermo Fisher Scientific cartridges marked such that the 400-mesh bars were in line with the bottom mark and the 100-mesh bars were in line with the side markings. This grid was loaded into the FIB shuttle, and 20-μm × 10-μm × 10-μm copper block adaptors were prepared as described above.

During the preparation of the receiver grid, roughly 5-μm-deep line patterns were milled across the horizontal 400-mesh grid bars in trench-milling orientation to divide the rectangles approximately one-third and two-thirds. These marker lines are intended to guide the alignment of the extracted volume by providing a reference visible in both the electron and ion beams during the attachment process.

**Lift-out, sectioning and milling procedure.** Lift-out site preparation for double-sided attachment was performed on an Aquilos 1 FIB–SEM instrument (Thermo Fisher Scientific) equipped with a METEOR in-chamber fluorescence light microscope (Delmic)[52]. The GIS was employed for 90 s at 27 °C and a stage working distance set to 10.6 mm to apply a layer of protective metal–organic platinum. The METEOR in-chamber microscope was used in conjunction with previously acquired cryo-confocal data to acquire fluorescence micrographs using Odemis version 3.2.1 and thus localize the larva of interest to produce an extraction volume of 180 μm × 42 μm × 25 μm. A width of 40 μm was necessary to span the space between the grid bars of a 100/400 rectangular mesh grid. A slight excess of width of the extraction volume

is preferable, as a too narrow block will not allow for double-sided attachment, while excess material can be ablated during sectioning.

The lift-out and receiver grid were subsequently transferred into an Aquilos 2 FIB–SEM instrument (Thermo Fisher Scientific) equipped with an EasyLift system. Lift-out was otherwise performed as stated in the section for single-sided attachment. In brief, the volume was attached to the needle via the copper adaptor, the volume was released, the needle was retracted, and the shuttle was returned to the lamella-milling orientation at the receiver grid. The extracted volume was reinserted, and its lower edge was positioned in between two grid bars. In the case when the volume was slightly too wide, material was ablated from its sides as necessary. To allow for proper attachment, the volume's lower edge was aligned to the reference line on the receiver grid in both the electron and ion beam images, and the volume was attached on both sides by redeposition from the grid bars. To achieve this, single-pass cross-section patterns were placed on the grid bars above the marker lines in close proximity to the interface between the grid bars and the extracted volume, while milling was directed away from the volume (Extended Data Fig. 10c). Once attached, the EasyLift system was maneuvered up in the z direction by 50–100 nm to create a small amount of strain. Subsequently, a section was released from the extracted volume by milling a line pattern at 4 μm above the volume's lower edge. This process was iterated until the entire extracted volume had been sectioned. The parameters for the different steps are given in (Table 1). Pattern files are provided (Supplementary Data 1).

### Fine milling of lift-out sections

For both single-sided and double-sided attachments, the initial 4-μm sections were thinned in two steps: rough milling and fine milling. Rough milling was performed using regular cross-section patterns at a beam current of 1 nA to a thickness of 1.5 μm, 0.5 nA to 1.2-μm lamella thickness and 0.3 nA to 0.8-μm lamella thickness. After rough milling all sections, fine milling was performed using regular rectangle patterns at a beam current of 0.1 nA to 0.4 μm and 50 pA to the final thickness. Over-tilting and under-tilting by up to 1° was used for beam-convergence compensation. The double-sided attached lamellae were sputter coated with platinum after fine milling for 4 s at a chamber pressure of 0.20 mbar and a current of 15.0 mA.

### Serial Lift-Out from high-pressure-frozen *D. melanogaster* egg chambers

Lift-out experiments of high-pressure-frozen *D. melanogaster* egg chambers were performed with a Scios FIB–SEM instrument (Thermo Fisher Scientific) equipped with an EasyLift needle. The preparation of the lift-out volume was described previously[18]. In brief, an approximately 20-μm × 20-μm volume of target material was prepared using regular cross-section patterns in trench-milling orientation in a horseshoe-like shape. The trenches were ~20 μm wide, except for the region that

needed to be accessed by the lift-out needle, which was ~35 μm wide. The extraction volume was undercut at a stage tilt of 45° or the highest tilt reachable for the specific position. This preparation leaves the extraction volume attached to the bulk material on a single side. Subsequently, the procedure for single-sided attachment was performed as described above. Sectioning was performed in increments of 1–3 μm using a line pattern milled at a beam current of 0.3 nA. After section preparation, fine milling was performed by sequentially decreasing the beam current: 0.3 nA to 800 nm, 0.1 nA to 500 nm, 50 pA to 350 nm and 30 pA to <300 nm. The final step was judged by loss of contrast in SEM imaging at an acceleration voltage of 3 kV and a beam current of 13 pA.

### Lamella preparation by the 'waffle' milling method

For preparation of lamellae directly on the high-pressure-frozen grid, vitrified as described above, we followed a similar workflow published as the 'waffle' method[21]. The grid was coated with a metal–organic platinum layer by GIS deposition for 20 s at a stage working distance of 10.6 mm. Milling was performed on a gallium FIB–SEM Aquilos 2 (Thermo Fisher Scientific) instrument. Initial trenches were milled in trench-milling orientation at a beam current of 3 nA using regular cross-section patterns. To avoid damaging the region of interest by milling at high beam currents, 2 μm of buffer distance was left around the region of interest. Trenches were extended to 15 μm on the backside of the subsequent lamella and 30 μm on the front side. In the next step, trenches were extended 1 μm closer to the region of interest at a beam current of 1 nA, and, in the last step, the final dimension of the lamella region was defined by milling at 0.5 nA. After trench milling, another layer of protective metal–organic platinum was deposited on the sample (deposition time, 20 s; stage working distance, 10.6 mm; three consecutive times). The stage was rotated into lamella-milling orientation, and a notch was milled with line patterns at 0.3 nA as previously described[21]. The preparation of the final lamella was carried out by removing material above and below of the region of interest at a beam current of 0.3 nA until a final lamella thickness of 2 μm was reached. In sequential steps of decreasing beam current (to 0.8-μm thickness at a beam current of 0.1 nA, to 0.4 μm at 50 pA), the remnant material was ablated. The lamella was polished to a final thickness of about 200–250 nm at a beam current of 30 pA.

### Transmission electron microscopy data acquisition

TEM data acquisition was performed on a Titan Krios G4 instrument at 300 kV equipped with a Selectris X energy filter and a Falcon 4i camera (Thermo Fisher Scientific). Lamella-overview montages were recorded by stage-driven tiling at a nominal magnification of ×11,500 (pixel size, 2.127 nm). Tomograms were recorded using the Tomo5 software package (version 5.12.0, Thermo Fisher Scientific) and the EER file format.

Two acquisition strategies were deployed. For tomograms of the single-sided attachment and 'waffle' preparation, tilt series were acquired at a nominal magnification of ×42,000, resulting in a pixel size at the sample of 2.93 Å. A dose-symmetric tilt scheme was used with an angular increment of 2°, a dose of 2 e⁻ Å⁻² per tilt and a target defocus of −4 to −5.5 μm. Tilt series were collected in a tilt range of −70° to 50° due to the lamella pre-tilt and with a total dose of 120 e⁻ Å⁻².

For the tomograms collected of the double-sided-attachment lamellae that were subsequently used for subtomogram analysis, tilt series were acquired at a nominal magnification of ×64,000, resulting in a pixel size of 1.89 Å. Data were collected using a dose-symmetric tilt scheme with an angular increment of 3°, a dose of 3.23 e⁻ Å⁻² per tilt and a target defocus range of −1 to −4 μm. Angles of −70° to 50° were acquired, resulting in a total dose of 132 e⁻ Å⁻².

### Tomogram reconstruction, visualization and subtomogram analysis for 80S ribosomes

Data were processed using the Tomoman version 0.7 pipeline (https://github.com/williamnwan/TOMOMAN). Fourteen frames with a dose of 0.23 e⁻ Å⁻² per frame were rendered from the EER files. These were used for motion correction in MotionCor2 version 1.4.7 (ref. 53) and CTF estimation with CTFFIND4 version 4.14 (ref. 54). Bad tilts were removed after manual inspection using the Tomoman script. Dose weighting was performed at 3.23 e⁻ Å⁻² per tilt using either Tomoman or Warp. For denoising, tilt series were separated into odd and even tilts during motion correction, and the resulting stacks were processed using Cryo-CARE[55]. Tomogram reconstructions for visualization were done with IMOD version 4.12.32 (refs. 56,57). Tilt series were aligned with AreTomo version 1.3.3. CTF-corrected tomograms for template matching were reconstructed with IMOD at bin8, resulting in a pixel size of 15.6 Å.

Initial template matching was performed in STOPGAP version 0.7 (ref. 58) on a subset of 70 tomograms at bin8 using PDB 4V4B as a reference filtered to 35 Å (ref. 59). A total of 16,420 particles were extracted and aligned in STOPGAP to generate the *C. elegans* 80S ribosome template. The template was subsequently used to repeat template matching on 200 tomograms at bin8. A total of 65,451 particles were extracted and cleaned by projecting the subtomogram along the z axis and subsequent 2D classification in RELION version 4.0 (ref. 60). The remaining dataset contained 37,026 particles. The retained particles were reprocessed in Warp version 1.0.9 (ref. 61) and cleaned to remove particles with inadequate CTF resolution and astigmatism. The resulting 35,350 particles were extracted with a pixel size of 2.98 Å and a box size of 160 pixels. RELION version 3.0 with a spherical mask 340 Å in radius was used to align the subtomograms. Finally, the particles were imported into M version 1.0.9 (ref. 62), and geometric and CTF parameters were sequentially refined. Corrected subtomograms were extracted from M and classified in RELION 3.0. Two rounds of classification with a spherical mask 340 Å in diameter resulted in 8,256 particles being removed. The second step was a focused classification with a spherical mask around the A–P–E site. The resulting five classes were combined into two classes according to small subunit rotation. Each of these two merged classes was subjected to another round of 3D classification with a mask around the elongation factor-binding site. The final classes were manually pooled according to structural similarity, yielding five classes with 2,175, 20,638, 877, 1,119 and 2,285 particles, of which the class with 877 particles was neglected. Segmentation of the elongation factor and tRNA was carried out with the Segger tool in Chimera. ChimeraX version 1.3 was used for visualization[63].

### Subtomogram averaging of microtubules

Microtubules were traced manually in IMOD using bin8 tomograms. Initial coordinates of 108 filaments were sampled every 10 Å along the filament centerline, and 20,220 particles were extracted at bin4. Subsequent processing was performed in STOPGAP version 0.7 unless otherwise stated. Due to in-plane randomization, the starting reference resembled a featureless tube, and initial alignment was carried out for each filament independently. The protofilament number and polarity was inferred from central z projections of the averages. As most filaments contained 11 protofilaments, filaments containing 15 protofilaments were discarded from further processing. These 15-protofilament microtubules originated from cells that are likely touch-receptor cells. Furthermore, microtubules with an average inclination angle above 60° with respect to the xy plane were excluded from further processing due to missing wedge effects, resulting in 38 filaments used for further processing.

To confirm the polarity for the remaining filaments, a multi-reference-based approach, as previously described[41], was employed, using two microtubules from the previous alignment as reference where the pseudohelical pattern was clearly visible in 3D. Subsequently, 8,903 particles from 28 filaments were aligned and averaged at bin4, followed by splitting the dataset into two halfsets and final refinement at bin2. Based on this structural information,

filaments were oversampled again to focus alignment on the tubulin subunits. A tubular grid was sampled with a diameter of 18 nm, and distances of 5 nm along the x axis and 4 nm along the z axis were over-sampled 4×, resulting in 108,372 particles from 28 filaments. Particles were extracted at bin4, and alignment was performed while limiting the shifts to 25 and 20 Å, respectively. Filaments for which a helical pattern did not emerge were removed, and distance cleaning was conducted for the remaining filaments, leading to 20,516 particles from 24 filaments, which were split into two halfsets and averaged at bin2. Particles were then reprocessed in Warp version 1.0.9 and aligned in RELION version 3.1 for one round at bin2 using a box size of 64 pixels. The resulting average was exported to M version 1.0.9 for one round of geometric multiparticle refinement. After removing duplicate particles from the previous step, a final refinement was performed in RELION version 3.1.

### Statistics and reproducibility

For this study, four Serial Lift-Out experiments and one on-grid 'waffle' lamella-milling experiment were performed as indicated in the figure legends. Experiment 1 was a single-sided attachment run on a 'waffle' sample containing L1 larvae of *C. elegans* strain AM140. Experiment 2 was a double-sided attachment run on a 'waffle' sample containing L1 larvae of *C. elegans* strain NK2476. Experiment 3 was a double-sided attachment run on a 'waffle' sample containing L1 larvae of *C. elegans* strain NK2476. Experiment 4 was a single-sided attachment run on *D. melanogaster* egg chambers from a strain expressing endogenously tagged GFP–NUP358. Experiment 5 used an on-grid lamella, milled from a 'waffle' sample of L1 larvae from *C. elegans* strain AM140.

Statistical analysis of lamella thickness was exclusively performed for the three *C. elegans* Serial Lift-Out experiments (experiments 1–3), as all subsequent subtomogram analysis was based on the data collected during these experiments. The Serial Lift-Out experiment performed on *D. melanogaster* egg chambers was included to show that the method can be applied to samples frozen directly in HPF carriers as well. The on-grid lamella was included to show the difference in sectioning geometry between lift-out and on-grid experiments.

Serial Lift-Out has been replicated in the Plitzko, Briggs and Baumeister departments of the Max Planck Institute of Biochemistry several tens of times with different users. Of these replicates, over 20 experiments have been performed with *C. elegans*. A broad range of other samples have been subjected to Serial Lift-Out, including moss, entire *C. elegans* worms at different developmental stages, *D. melanogaster* and other invertebrate tissues and adherent human cell lines. In these experiments, incomplete vitrification by HPF rather than lamella preparation was the limiting factor for the collection of tilt series. Double-sided attachment has primarily been used for these experiments, as it yielded more uniform lamella thicknesses.

### Reporting summary

Further information on research design is available in the Nature Portfolio Reporting Summary linked to this article.

## Data availability

Tomograms have been deposited in the Electron Microscopy Data Bank under accession codes EMD-17246, EMD-17247, EMD-17248 and EMD-18186. Subtomogram averages have been uploaded under accession numbers EMD-17241, EMD-17242, EMD-17243, EMD-17244, EMD-17245 and EMD-18187 and will be released upon publication. Source data are provided with this paper.

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

## Acknowledgements

We thank J. Wagner, A. Prajica, P.S. Erdmann, J. Brenner, M. Li, A. Rigort, B. Hampoelz, M. Beck and R. Fässler for invaluable input and support. We thank the Caenorhabditis Genetics Center and the Bloomington Drosophila Stock Center for providing *C. elegans* and *D. melanogaster* strains, respectively. This study used the infrastructure of the Department of Cell and Virus Structure at the MPI of Biochemistry. S.K. was supported by the International Max Planck Research School for Molecular and Cellular Life Sciences. This work was funded by the Max Planck Society (J.M.P. and J.A.G. Briggs).

## Author contributions

O.H.S., C.J.O.K., S.K. and J.M.P. conceptualized the work. O.H.S., C.J.O.K., S.K. and D.P.K. prepared samples and performed lift-out experiments. O.H.S., C.J.O.K., S.K., D.R.M. and D.P.K. collected

cryo-ET data. O.H.S., C.J.O.K., F.B., J.S. and S.K. analyzed and visualized data. C.T. developed the copper block adaptor. The initial draft was written by S.K., and review and editing of the paper were carried out by O.H.S., C.J.O.K., S.K. and J.M.P. with input from all authors. Funding was acquired by J.M.P.

## Funding

## Competing interests

J.M.P. holds a position on the advisory board of Thermo Fisher Scientific. C.T. is an employee of Thermo Fisher Scientific and holds the patent for the copper block attachment method (EP 4 207 243 A1). The other authors declare no competing interests.

## Additional information

**Extended data** is available for this paper at https://doi.org/10.1038/s41592-023-02113-5.

**Correspondence and requests for materials** should be addressed to Sven Klumpe or Jürgen M. Plitzko.

**Peer review information** Peer reviewer reports are available.

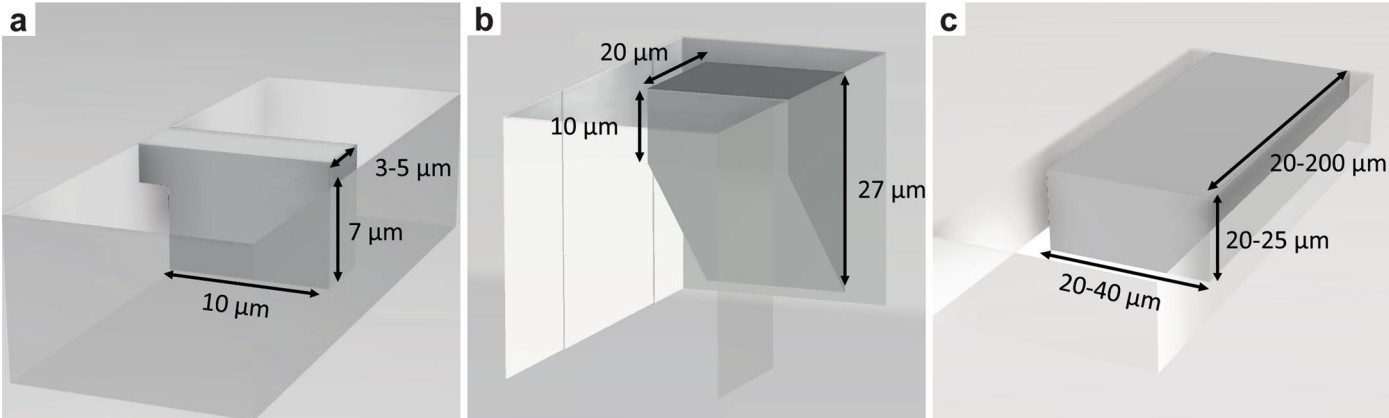

**Extended Data Fig. 1 | Schematics of lift-out geometries.** Geometries of the extracted volume for different lift-out approaches are illustrated. Extraction volumes are depicted in dark grey, with the empty volume around them indicating trenches milled prior to the lift-out procedure. **a**, Lamella lift-out prepares a thin section with a small amount of excess material around the final lamella volume, dispatched from the bulk by a J-cut. **b**, Lift-out of larger volumes from thick HPF samples requires detachment of the material from the bulk by an undercut. This lift-out geometry for thick HPF preparations generally limits the extraction volume to approximately 20–30 μm in height, dependent on the angle of the undercut. **c**, Lift-out from 'waffle'-type samples allows the creation of large extraction volumes. As no undercut is required, the entire thickness of the sample can be extracted.

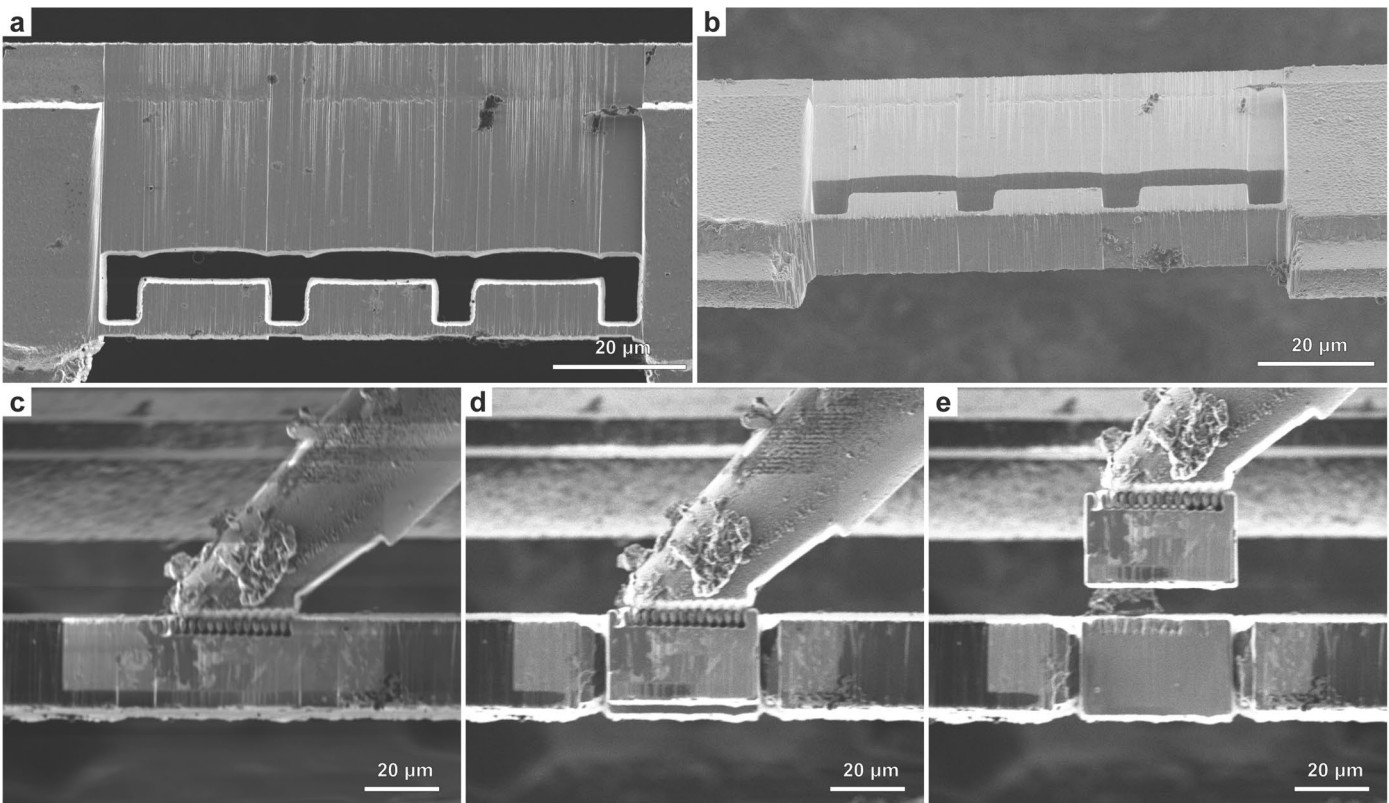

**Extended Data Fig. 2 | Copper block adapter preparation and attachment.** The grid bars of the receiver grid can be used to prepare copper blocks that serve as adapters for the attachment of the extracted volume to the tungsten needle. **a**, FIB top view and **b**, SEM side view of a grid bar, with copper blocks prepared for extraction. Three blocks were milled into a bar of a copper 100 mesh grid (trench milling orientation). **c-e**, FIB images at lamella milling orientation. The tungsten needle of the micromanipulator was flattened to increase the copper block contact area and the block is attached by redeposition milling (**c**). The copper block is subsequently milled free from the grid bar (**d**) and lifted out (**e**). All steps were performed using a 45° pre-tilt shuttle.

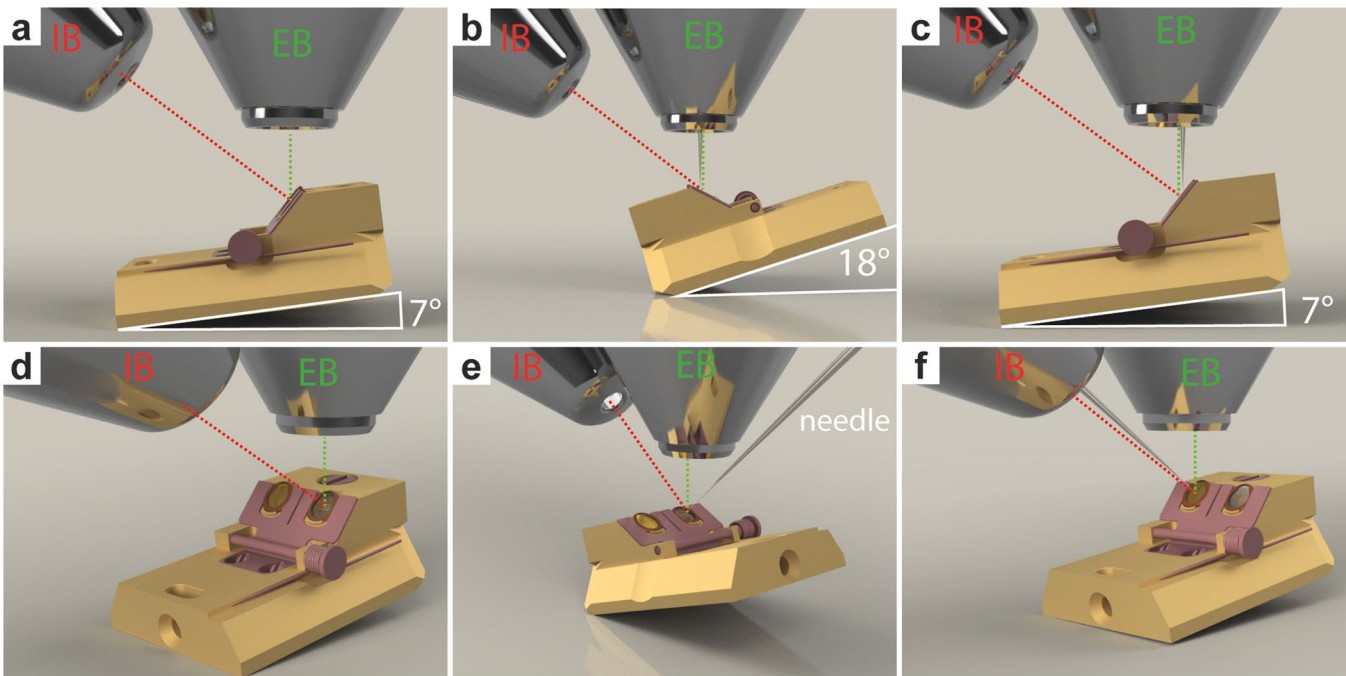

**Extended Data Fig. 3 | Stage orientations for Serial Lift-Out with a 45° pre−tilted shuttle. a**, **d**, Copper block preparation and trench milling are performed at 7° tilt angle and 180° relative stage rotation (trench milling orientation) **b**,**e**, Lift-out of the copper block is performed at 18° tilt angle and 0° relative rotation (lamella milling orientation). The same orientation is used during the attachment of Serial Lift-Out slices to the receiver grid and lamella thinning. **c**,**f**, Lift-out to obtain sections perpendicular to the grid plane is performed at trench milling orientation. Panels **d-f** are oblique front views of the corresponding side views shown in **a-c**. From this perspective, the HPF sample is located in the left shuttle position and the receiver grid is located the right shuttle position. Panels **b**,**c**,**e**,**f** show the EasyLift-needle inserted. EB and IB indicate the column of the ion beam and electron beam, respectively.

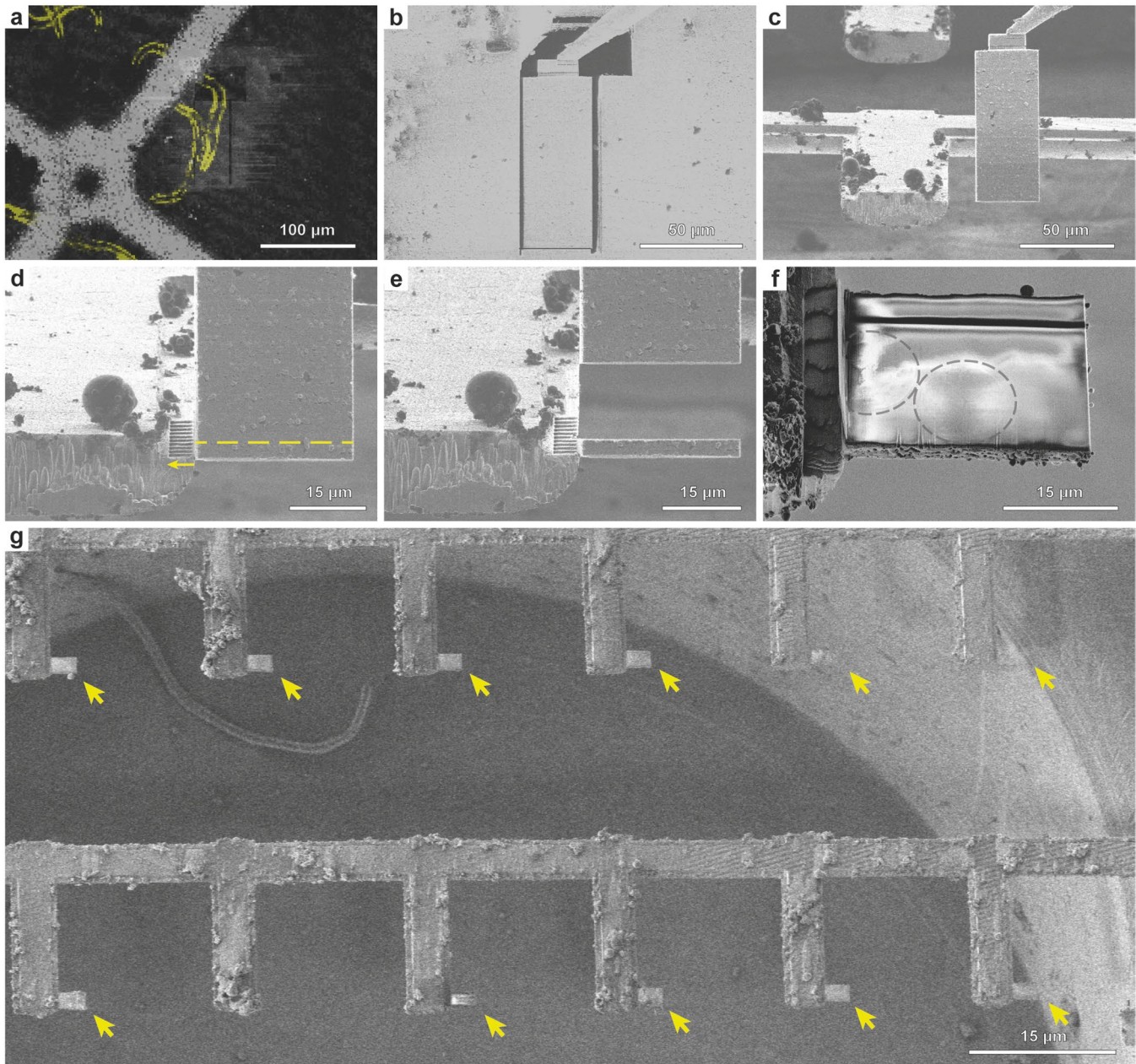

**Extended Data Fig. 4 | A workflow for single−sided attachment Serial Lift-Out. a**, FIB image of the extraction site with overlaid correlated fluorescence data (yellow) indicating the larva being targeted (trench milling orientation). **b**, The extraction volume is attached to the EasyLift needle using redeposition from the copper adapter. The release cut, milled with a line patten, is noticeable at the base of the extraction volume (trench milling orientation). **c**, FIB image of the extracted volume being lowered to the attachment position adjacent to a pin. For attachment, the lower front edge of the volume is aligned to the corner of the pin. **d**, Attachment using redeposition from the pin (yellow arrow indicates milling direction), followed by line pattern milling releasing the section of a desired thickness (dashed yellow line). **e**, The resulting section is depicted as the remaining extracted volume is retracted. **f**, SEM image of a typical section after release from the extraction volume. Note the faint pattern of worm cross-sections discernible (grey dashed lines). **g**, SEM image of the resulting receiver grid after a session of single−sided attachment Serial Lift-Out. Arrows indicate the 12 sections obtained. Supplementary Movie 2 summarizes the process.

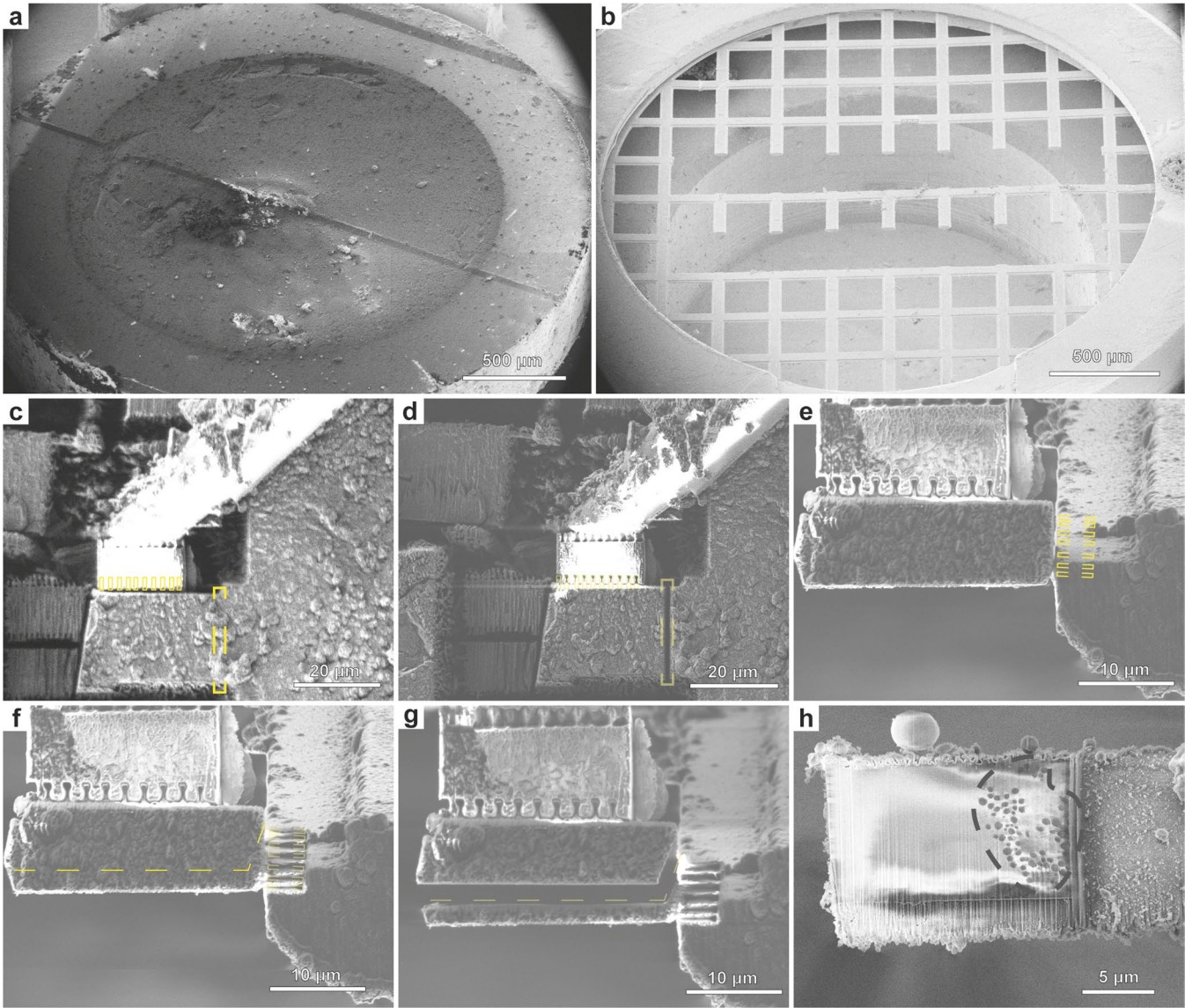

**Extended Data Fig. 5 | Voluminous HPF samples by single–sided attachment Serial Lift-Out.** Serial Lift-Out experiment performed on *D. melanogaster* egg chambers vitrified in HPF sample carriers. **a**, HPF sample carrier planed by diamond knife trimming at 45° in a cryo-ultramicrotome to access regions deep within the egg chamber. **b**, Receiver grid as prepared by FIB milling (Supplementary Fig. 2). **c**, FIB image of the target extraction volume before attachment to the needle system. Redeposition patterns are indicated with yellow boxes. **d**, The extracted volume is released from the bulk by milling a regular cross-section pattern (dashed yellow box) and transferred to the receiver grid. **e**, FIB image of the extracted volume being lowered to the attachment position adjacent to a pin. For attachment, the lower front edge of the volume is aligned to the corner of the pin. Redeposition patterns are indicated with yellow boxes. **f**, The extracted volume is attached to the pin by redeposition from the grid bars (yellow dashed boxes). The section is separated from the remaining volume by line pattern milling (dashed line). **g**, The remaining volume is retracted for successive rounds of sectioning. The sliced section remains attached to the pin. **h**, SEM image view of the section shown in **g** after polishing the surface by low current milling. Note the clearly visible lipid droplets in the cytoplasmic region of the egg chamber nurse cell (grey dashed line). The region to the left, lacking lipid droplets, is the nucleoplasmic region that was targeted prior by fluorescence microscopy and confirmed by subsequent TEM imaging. **c**, **d**, Trench milling orientation. **e–h**, Lamella milling orientation.

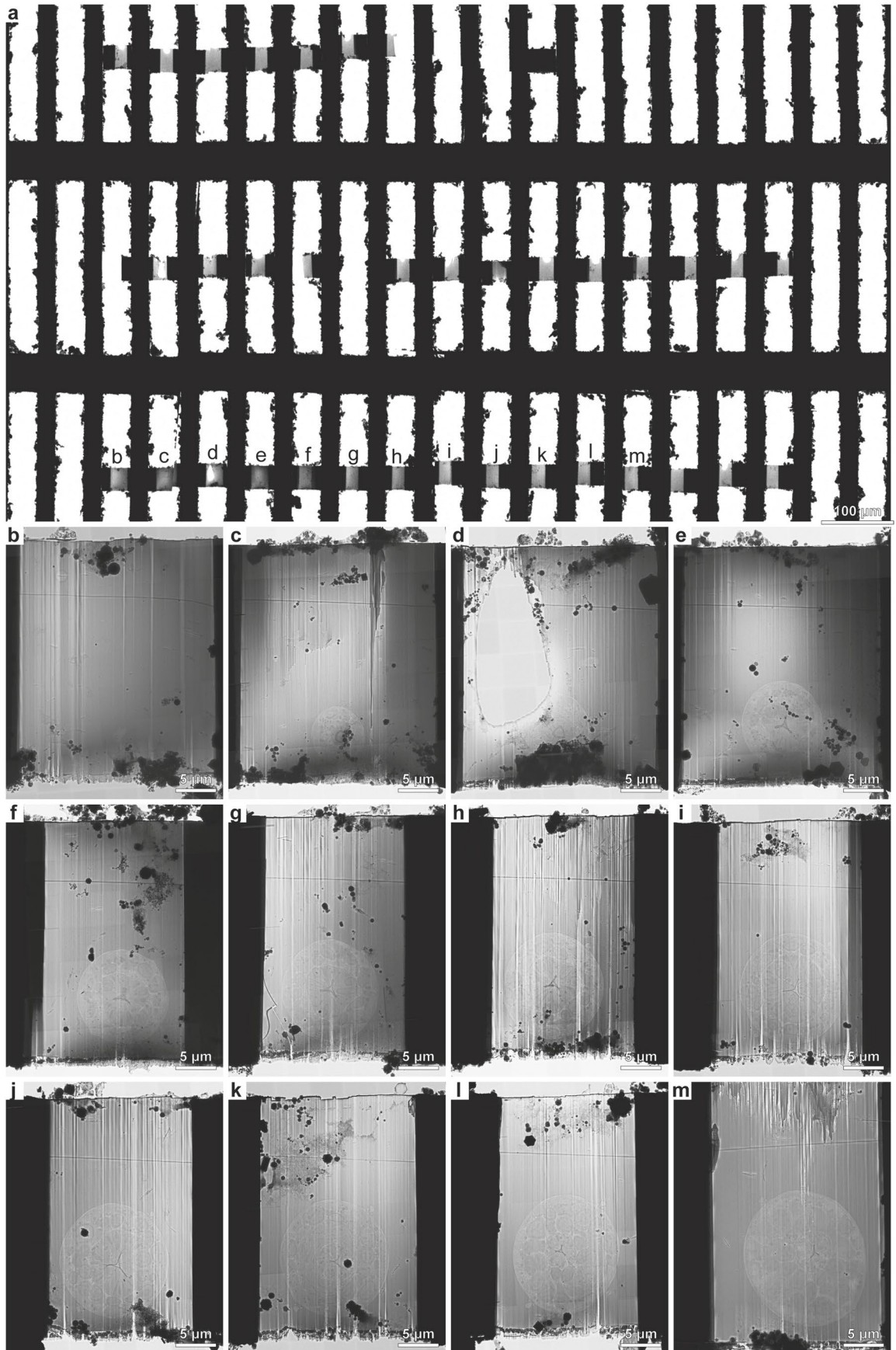

**Extended Data Fig. 6 | TEM overview of the grid derived from the double-sided attachment Serial Lift-Out experiment. a**, TEM low magnification overview image (magnification 125×) of the lamella region of the Serial Lift-Out receiver grid for double-sided attachment. **b-m**, Lamella overviews (magnification 11,500x) of the first 12 sections. In total, 40 sections were prepared, of which 39 were milled to lamella thickness. 33 were successfully transferred into the TEM. The letters in panel a indicate the corresponding higher magnification micrographs in panels **b-m**.

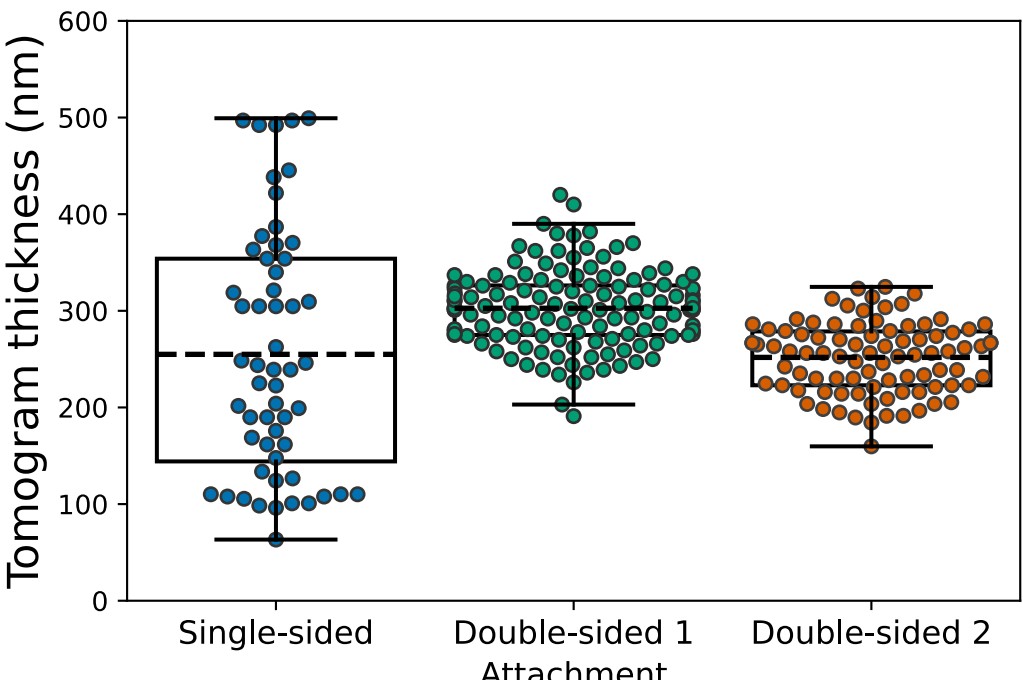

**Extended Data Fig. 7 | Tomogram thickness distribution for the lift-out experiments.** A box plot for the distribution of tomogram thickness measurements on single-sided and double-sided attachment Serial Lift-Out lamellae. To show the spread of thicknesses among technical replicates of the Serial Lift-Out method, we analyzed the thicknesses obtained from multiple independent experiments. All n = 56 tomograms recorded for the single-sided attachment experiment (experiment 1) and a random selection of n = 132 tomograms from the 1012 tilt series collected for the double-sided attachment 1(experiment 2) were analyzed. All n = 90 tilt series of double-sided attachment 2 (experiment 3) were analyzed. While the average tomogram thickness is lower for single-sided attachment (255 nm ± 126 nm, n = 56) than for double-sided attachment 1 (303 nm ± 40 nm, n = 132) and similar to double-sided attachment 2 (252 nm ± 36 nm, n = 90), the spread of thickness measurements is higher for single-sided attachment, likely due to increased bending of the free-standing lamellae during milling. The box indicates the 25th and 75th percentile, the dashed line the mean of all measurements, and the whiskers indicate the observations within the 1.5x interquartile range. Experiment numbering is given in the Statistics and Reproducibility section.

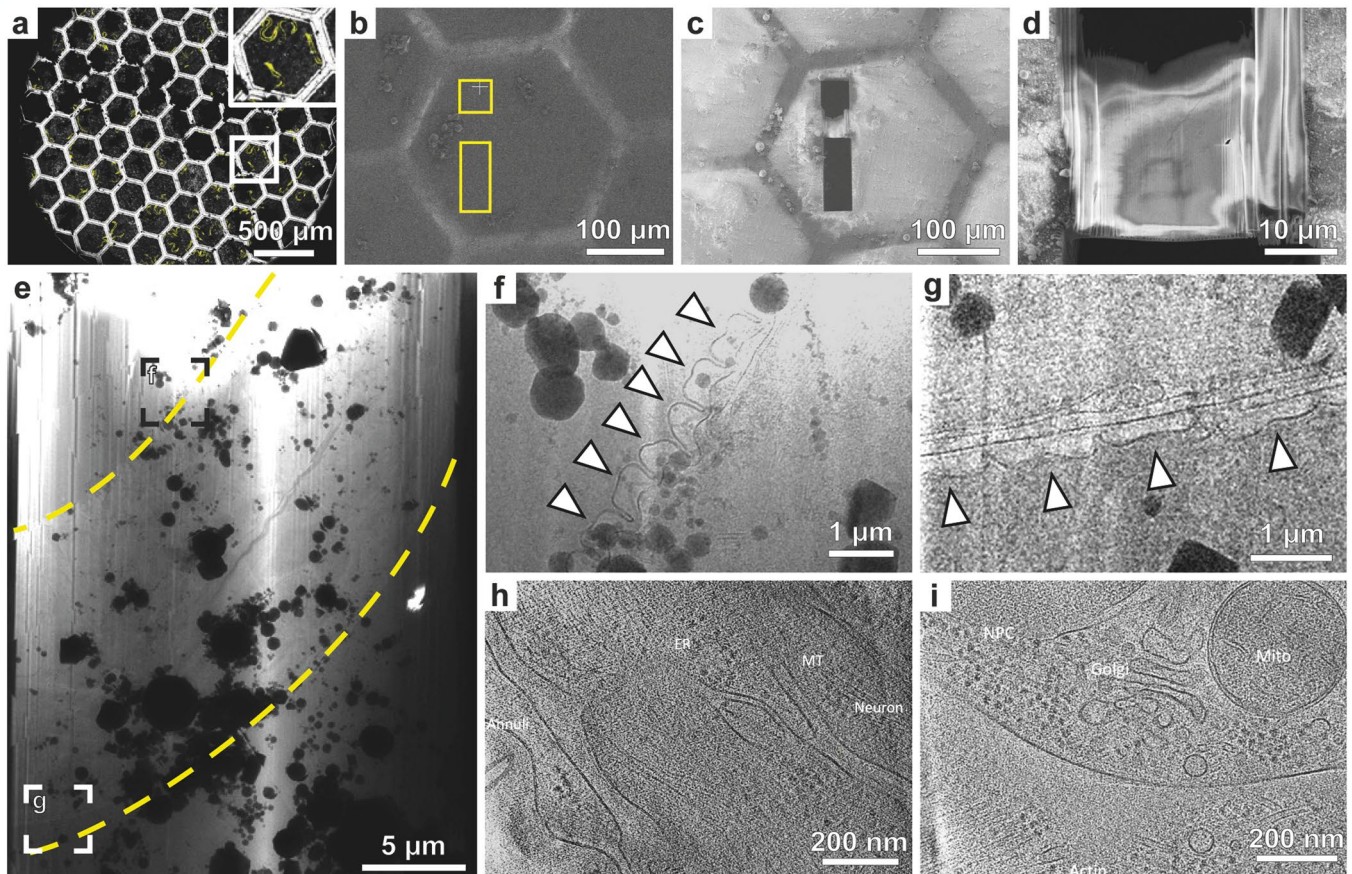

**Extended Data Fig. 8 | 'Waffle' lamella milling of a *C. elegans* L1 larva.**
**a**, Fluorescence (yellow) and reflection (white) channel confocal micrograph of the grid used for 'waffle' milling. The inset highlights the mesh in which the larva was targeted. **b**, **c**, FIB view images of the mesh shown in inset **a** pre- (**b**) and post- milling (**c**). Trenches were milled at the trench milling orientation (sample surface perpendicular to the ion beam). Yellow rectangles in **b** indicate the trench milling locations. **d**, SEM overview of the final lamella. **e**, TEM overview of the lamella. Yellow dashed lines indicate the lateral outline of the larva. **f**, **g**, Structural difference between the body wall regions of the contracted and relaxed sides of the worm freely thrashing before vitrification. This difference can be clearly seen through the morphological change in the annuli (arrowheads). **h**, **i**, Tomograms recorded on the lamella shown in **e** revealing biological features as obtained from the longitudinal section. ER: endoplasmic reticulum, MT: microtubule, NPC: nuclear pore complex, Mito: mitochondria.

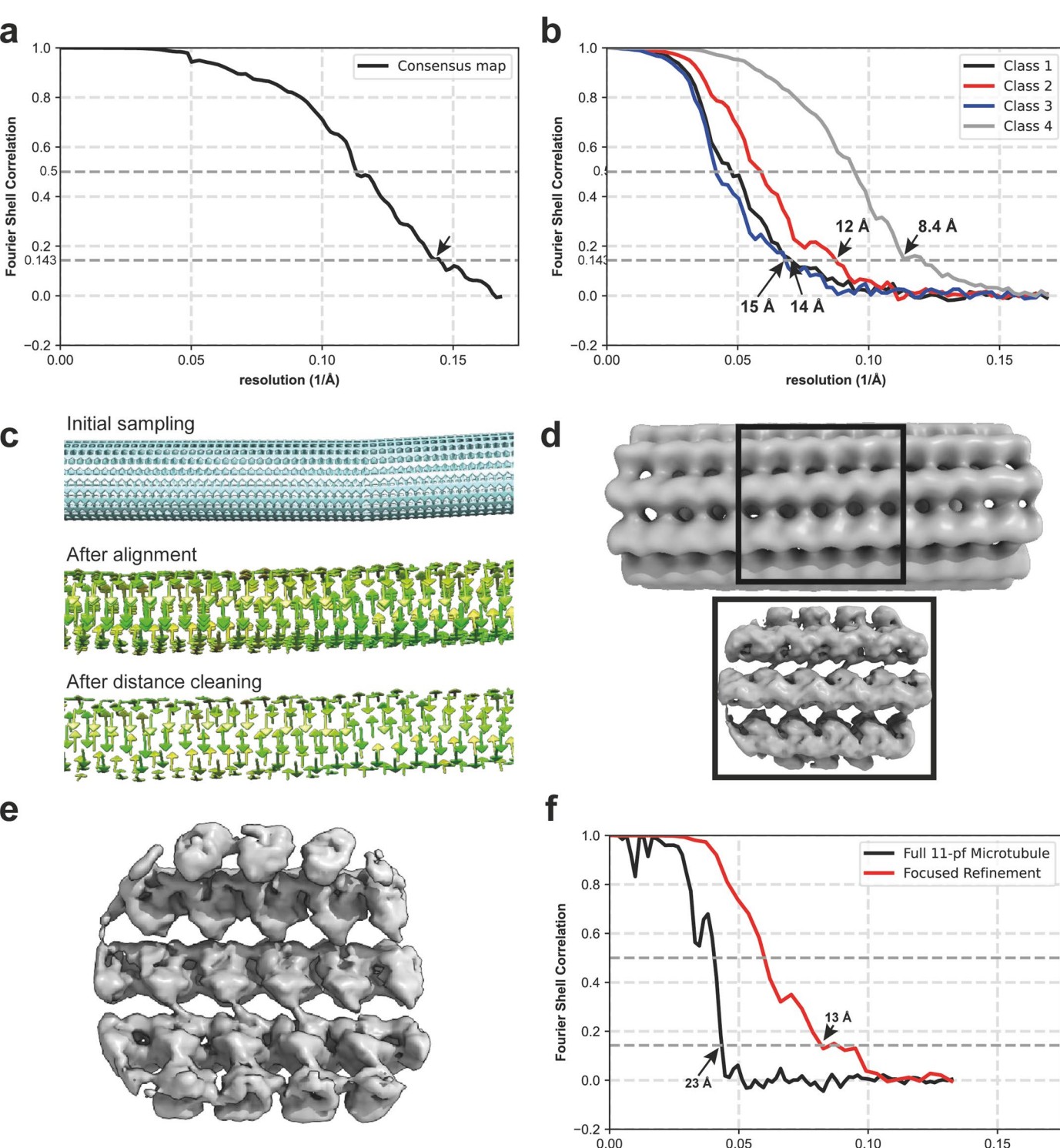

**Extended Data Fig. 9 | Fourier shell correlation (FSC) analysis and subtomogram averages of the 80S ribosome and a microtubule subregion.** **a**, FSC curves of the consensus average and **b**, the four different translational state classes of the *C. elegans* ribosome. **c**, Particle positions and orientations for focused alignment during initial sampling, after alignment and after distance cleaning visualized by the Place Object plugin in UCSF Chimera. **d**, The 23 Å structure of the full 11-protofilament microtubule. The highlighted area indicates the subregion obtained from focused refinement. **e**, The 13 Å refined structure of the 11-protofilament microtubule, visualized from the luminal side. **f**, FSC curves of the full (black) and focused (red) refinement of the 11-protofilament microtubule.

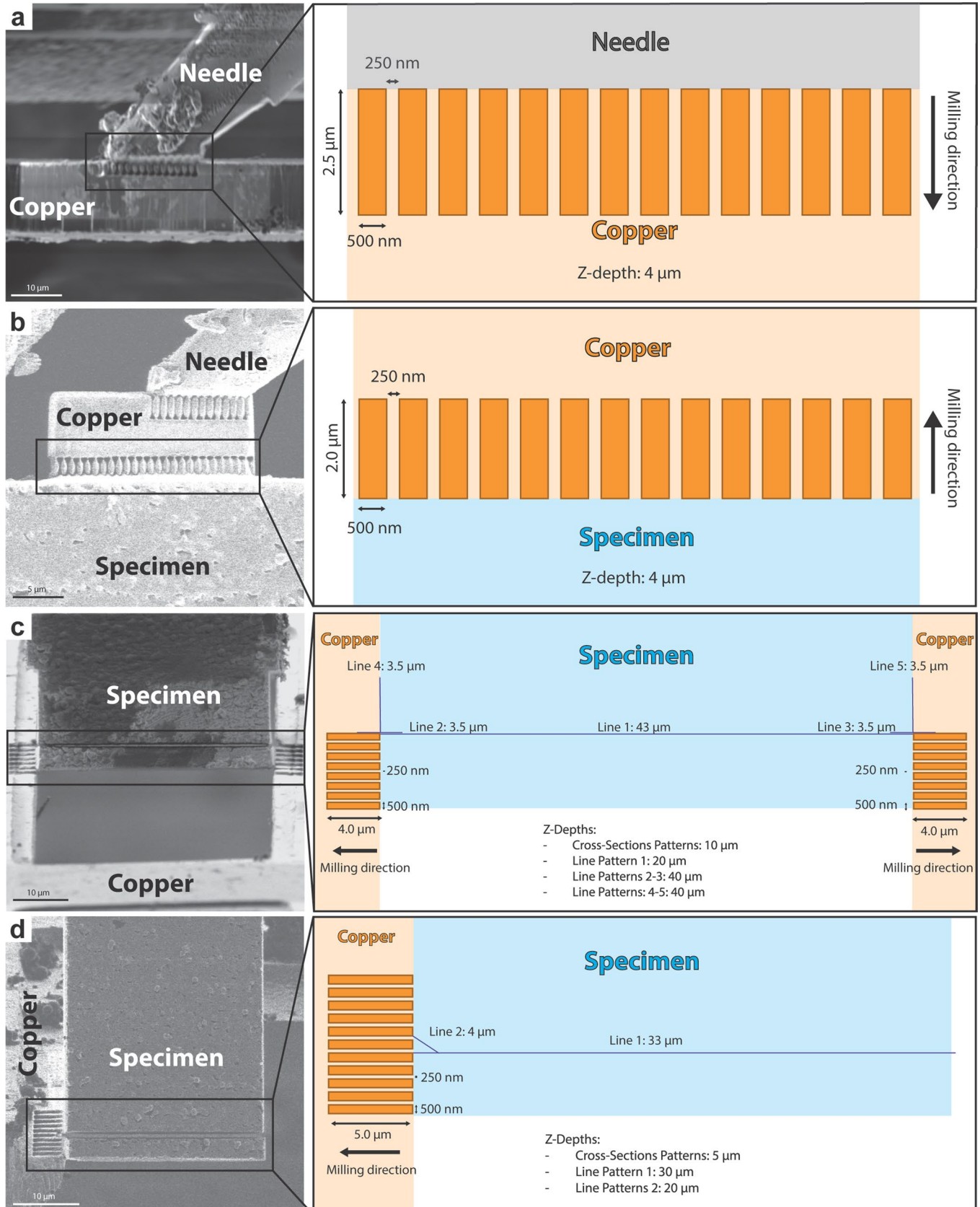

**Extended Data Fig. 10 | See next page for caption.**

**Extended Data Fig. 10 | Serial Lift-Out milling pattern dimensions.**
A schematic of the milling patterns used for double-sided and single-sided attachment Serial Lift-Out. Orange rectangles represent single-pass cross-section patterns used for attachment, while blue lines indicate line patterns used to section. Note that the exact dimensions of the redeposition patterns may need to be adjusted for different instruments. **a**, Copper block redeposition attachment to the EasyLift system. **b**, Copper block redeposition attachment to the extracted volume. **c**, Double-sided attachment: Redeposition attachment from the grid bars to the specimen, followed by section release by line pattern milling. **d**, Single-sided attachment: Redeposition attachment from the pin to the specimen, followed by section release by line pattern milling. Pattern files for Thermo Fisher Scientific FIB-SEM instruments are provided in Supplementary Data 1.

# Reporting Summary

## Statistics

For all statistical analyses, confirm that the following items are present in the figure legend, table legend, main text, or Methods section.

| n/a | Confirmed | |
|---|---|---|
| ☐ | ☒ | The exact sample size (*n*) for each experimental group/condition, given as a discrete number and unit of measurement |
| ☐ | ☒ | A statement on whether measurements were taken from distinct samples or whether the same sample was measured repeatedly |
| ☒ | ☐ | The statistical test(s) used AND whether they are one- or two-sided *Only common tests should be described solely by name; describe more complex techniques in the Methods section.* |
| ☒ | ☐ | A description of all covariates tested |
| ☒ | ☐ | A description of any assumptions or corrections, such as tests of normality and adjustment for multiple comparisons |
| ☐ | ☒ | A full description of the statistical parameters including central tendency (e.g. means) or other basic estimates (e.g. regression coefficient) AND variation (e.g. standard deviation) or associated estimates of uncertainty (e.g. confidence intervals) |
| ☒ | ☐ | For null hypothesis testing, the test statistic (e.g. *F*, *t*, *r*) with confidence intervals, effect sizes, degrees of freedom and *P* value noted *Give P values as exact values whenever suitable.* |
| ☒ | ☐ | For Bayesian analysis, information on the choice of priors and Markov chain Monte Carlo settings |
| ☒ | ☐ | For hierarchical and complex designs, identification of the appropriate level for tests and full reporting of outcomes |
| ☒ | ☐ | Estimates of effect sizes (e.g. Cohen's *d*, Pearson's *r*), indicating how they were calculated |

*Our web collection on statistics for biologists contains articles on many of the points above.*

## Software and code

Policy information about availability of computer code

| Data collection | FIB/SEM data: XT v 20.1.1<br>Confocal cryo fluorescence data: LAS X 3.5.5.19976<br>Wide field cryo fluorescence data: Odemis 3.2.1<br>Tilt series acquisition: Tomo5 5.12.0 |
|---|---|
| Data analysis | Overview processing: ImageJ 2.9.0 + TrackEM plugin<br>Fluorescence data: MAPS 3.14<br>Tilt series/tomograms: IMOD 4.12.32, AreTomo 1.3.3, STOPGAP 0.7, TOMOMAN 0.7, Relion 4.0, Relion 3.0, Relion 3.1, WARP/M 1.0.9 , ChimeraX 1.3. MotionCor2 1.4.7, CTFFIND4 4.14 |

For manuscripts utilizing custom algorithms or software that are central to the research but not yet described in published literature, software must be made available to editors and reviewers. We strongly encourage code deposition in a community repository (e.g. GitHub). See the Nature Portfolio guidelines for submitting code & software for further information.

## Data

Policy information about availability of data

All manuscripts must include a data availability statement. This statement should provide the following information, where applicable:
- Accession codes, unique identifiers, or web links for publicly available datasets
- A description of any restrictions on data availability
- For clinical datasets or third party data, please ensure that the statement adheres to our policy

Tomograms have been deposited in the Electron Microscopy Data Bank (EMDB) under accession codes: EMD-17246, EMD-17247, EMD-17248, EMD-18186 Subtomogram averages have been uploaded under accession numbers: EMD-17241, EMD-17242, EMD-17243, EMD-17244, EMD-17245, EMD-18187, and will be released upon publication.

## Human research participants

Policy information about studies involving human research participants and Sex and Gender in Research.

| | |
|---|---|
| Reporting on sex and gender | N/A |
| Population characteristics | N/A |
| Recruitment | N/A |
| Ethics oversight | N/A |

Note that full information on the approval of the study protocol must also be provided in the manuscript.

# Field-specific reporting

Please select the one below that is the best fit for your research. If you are not sure, read the appropriate sections before making your selection.

☒ Life sciences    ☐ Behavioural & social sciences    ☐ Ecological, evolutionary & environmental sciences

For a reference copy of the document with all sections, see nature.com/documents/nr-reporting-summary-flat.pdf

# Life sciences study design

All studies must disclose on these points even when the disclosure is negative.

| | |
|---|---|
| Sample size | Sample sizes were not determined through statistical methods. For the three separate C. elegans Serial Lift-Out and single on-grid experiments, four individuals from separate strains (twice AM140, both from one 'waffle' grid, twice NK2476 from one 'waffle' grid) were used. One egg chamber of Drosophila melanogaster (GFP-Nup358 endogenously tagged) was used for the Serial Lift-Out experiment from high-pressure freezing planchettes. Descriptive statistics on tomogram thicknesses is based on either all tomograms (n=56 for Experiment 1, n=90 for Experiment 3) or a randomized subset (n=132 for Experiment 2) of reconstructed tomograms for experiment . |
| Data exclusions | Tomograms for subtomogram averaging but not for tomogram thickness measurement were excluded based on on poor reconstruction quality and excessive thickness. Particles were excluded based on classification during subtomogram averaging when not contributing to an improvement of the GSFSC criterion. |
| Replication | Serial Lift-Out has been replicated several tens of times on different sample types by the authors and other members of the Baumeister and Briggs departments at the MPI of Biochemistry (MPI of Biochemistry, Martinsried, Germany). The method was adopted in other labs including the labs of Stefan Raunser (MPI for Molecular Physiology, Dortmund, Germany), Philipp Erdmann (Human Technopole, Milan, Italy), and Julia Mahamid (EMBL Heidelberg, Germany). |
| Randomization | Individuals of C. elegans were selected solely based on the criterion of accessibility for lift-out, mainly influenced by body curvature, lateral position and orientation on the grid. Tomograms for suntomogram averageing were only selected based on lamella thickness and otherwise randomly from the whole dataset. |
| Blinding | Data collection and analysis were not blinded. |

# Reporting for specific materials, systems and methods

We require information from authors about some types of materials, experimental systems and methods used in many studies. Here, indicate whether each material, system or method listed is relevant to your study. If you are not sure if a list item applies to your research, read the appropriate section before selecting a response.

## Materials & experimental systems

| n/a | Involved in the study |
|---|---|
| ☒ | Antibodies |
| ☒ | Eukaryotic cell lines |
| ☒ | Palaeontology and archaeology |
| ☐ | ☒ Animals and other organisms |
| ☒ | Clinical data |
| ☒ | Dual use research of concern |

## Methods

| n/a | Involved in the study |
|---|---|
| ☒ | ChIP-seq |
| ☒ | Flow cytometry |
| ☒ | MRI-based neuroimaging |

## Animals and other research organisms

Policy information about studies involving animals; ARRIVE guidelines recommended for reporting animal research, and Sex and Gender in Research

| | |
|---|---|
| Laboratory animals | Caenorhabditis elegans, Drosophila melanogaster |
| Wild animals | No wild animals were used. |
| Reporting on sex | Hermaphrodites (C. elegans), female (D. melanogaster) |
| Field-collected samples | No field samples were used. |
| Ethics oversight | C. elegans and D. melanogaster do not require ethic approval for use in experiments. |

Note that full information on the approval of the study protocol must also be provided in the manuscript.

