## [Peer Review File · Nature Methods]

Peer Review Information

Manuscript Title: Serial Lift-Out – Sampling the Molecular Anatomy of Whole Organisms

Corresponding author name(s): Sven Klumpe, Juergen Plitzko

Editorial Notes: None

Reviewer Comments & Decisions:

Decision Letter, initial version:

Dear Sven,

Your Article, "Serial Lift-Out – Sampling the Molecular Anatomy of Whole Organisms", has now been seen by two reviewers. As you will see from their comments below, although the reviewers find your work of considerable potential interest, they have raised a few concerns. We are interested in the possibility of publishing your paper in Nature Methods, but would like to consider your response to these concerns before we reach a final decision on publication.

We therefore invite you to revise your manuscript. We ask that you address the few referee concerns, add the additional demonstration we discussed prior to review, and add a detailed Supplementary Protocol describing the workflow on a "typical" sample.

* include a point-by-point response to the reviewers and to any editorial suggestions

* please underline/highlight any additions to the text or areas with other significant changes to facilitate review of the revised manuscript

- * address the points listed described below to conform to our open science requirements
- * ensure it complies with our general format requirements as set out in our guide to authors at www.nature.com/naturemethods
- * resubmit all the necessary files electronically by using the link below to access your home page

[Redacted] This URL links to your confidential home page and associated information about manuscripts you may have submitted, or that you are reviewing for us. If you wish to forward this email to co-authors, please delete the link to your homepage.

We hope to receive your revised paper within two months. If you cannot send it within this time, please let us know. In this event, we will still be happy to reconsider your paper at a later date so long as nothing similar has been accepted for publication at Nature Methods or published elsewhere.

OPEN SCIENCE REQUIREMENTS

REPORTING SUMMARY AND EDITORIAL POLICY CHECKLISTS

Please note that these forms are dynamic 'smart pdfs' and must therefore be downloaded and completed in Adobe Reader. We will then flatten them for ease of use by the reviewers. If you would

like to reference the guidance text as you complete the template, please access these flattened versions at <http://www.nature.com/authors/policies/availability.html>.

DATA AVAILABILITY

We strongly encourage you to deposit all new data associated with the paper in a persistent repository where they can be freely and enduringly accessed. We recommend submitting the data to discipline-specific and community-recognized repositories; a list of repositories is provided here:

<http://www.nature.com/sdata/policies/repositories>

All novel DNA and RNA sequencing data, protein sequences, genetic polymorphisms, linked genotype and phenotype data, gene expression data, macromolecular structures, and proteomics data must be deposited in a publicly accessible database, and accession codes and associated hyperlinks must be provided in the “Data Availability” section.

Please include a “Data availability” subsection in the Online Methods. This section should inform readers about the availability of the data used to support the conclusions of your study, including accession codes to public repositories, references to source data that may be published alongside the paper, unique identifiers such as URLs to data repository entries, or data set DOIs, and any other statement about data availability. At a minimum, you should include the following statement: “The data that support the findings of this study are available from the corresponding author upon request”, describing which data is available upon request and mentioning any restrictions on availability. If DOIs are provided, please include these in the Reference list (authors, title, publisher (repository name), identifier, year). For more guidance on how to write this section please see:

<http://www.nature.com/authors/policies/data/data-availability-statements-data-citations.pdf>

CODE AVAILABILITY

Please include a “Code Availability” subsection in the Online Methods which details how your custom code is made available. Only in rare cases (where code is not central to the main conclusions of the paper) is the statement “available upon request” allowed (and reasons should be specified).

SUPPLEMENTARY PROTOCOL <--- PLEASE ADD

To help facilitate reproducibility and uptake of your method, we ask you to prepare a step-by-step Supplementary Protocol for the method described in this paper. We encourage authors to share their step-by-step experimental protocols on a protocol sharing platform of their choice and report the protocol DOI in the reference list. Nature Portfolio 's Protocol Exchange is a free-to-use and open resource for protocols; protocols deposited in Protocol Exchange are citable and can be linked from the published article. More details can found at www.nature.com/protocolexchange/about.

ORCID

Nature Methods is committed to improving transparency in authorship. As part of our efforts in this direction, we are now requesting that all authors identified as ‘corresponding author’ on published papers create and link their Open Researcher and Contributor Identifier (ORCID) with their account on the Manuscript Tracking System (MTS), prior to acceptance. This applies to primary research papers only. ORCID helps the scientific community achieve unambiguous attribution of all scholarly contributions. You can create and link your ORCID from the home page of the MTS by clicking on ‘Modify my Springer Nature account’. For more information please visit please visit www.springernature.com/orcid.

Sincerely,
Rita

Rita Strack, Ph.D.
Senior Editor
Nature Methods

Reviewers' Comments:

Reviewer #1:
Remarks to the Author:

Cryo-ET on multicellular model organisms and organoids of all kind have great potential for discoveries and drug testing but this will only happen after many method development obstacles have been addressed properly. Plasma focused ion beam technology showed recently great potential for real structural cell biology and the sub nanometer resolution in lamellae of intact cells.

Cryo-lift-out procedure of lamellae of multicellular tissues as discussed in the study has been problematic because of the low success rate. Not many groups invested significant effort to overcome all these major problems except a few pioneers in the field. The team of Jurgen Plitzko is clearly a great example of searching for novel methodology. This manuscript shows several milestones in method development to transform in a novel and creative way the boundaries of the lift-out for multicellular samples.

Lift-Out was also initiated by the Plitzko / Baumeister team a few years ago and this study of Serial Lift-Out is one of the next steps. This team shows the ability to section in increments of one to four micrometers. This process has two steps that contribute to specimen loss: sectioning (~300-500 nm) and lamella milling. Serial Lift-Out sections and tomograms have been mapped back into context using other sources of volumetric data as illustrated.

This work is very original and significant for the entire field of cryo-EM. The work is very well presented and each paragraph is easy to understand for an insider. I regard the data of high quality and the figures

are well presented. I support the validity, the reliability with the robust conclusions without any hesitation.

Minor comments:

The statement in the text: “While the former yields samples that are easily FIB-milled, the sample thickness that can reliably be vitrified is limited to roughly 10 μm” contradicts and earlier study of this group (Mahamid et al., in Science - Visualizing the molecular sociology at the HeLa cell nuclear periphery) where it was stated: “clearly, the centre of the cell undergoes incompletely vitrification due to the heat transfer ...”. As long as we have no strong quantitative data of reproducible measurement of thickness it is better not to overstate.

Line 392: an error in the sentence: This like likely due to the reduction of lamella ...

Line 551: an error in the sentence: The the GIS was ...

Reviewer #2:

Remarks to the Author:

This manuscript describes an important development that would allow the structural study of tissues by cryo-ET. The use of cryo-FIB as a tool for sample preparations of cells, is gaining momentum in the last several of years. It becomes a common practice to produce lamellae from individual vitrified cells. However, working with multi cellular specimen is not yet trivial and only few examples were previously demonstrated and published. To this end, and in order to develop a reproducible procedure, the authors developed an elegant strategy that facilitate cryo-ET study of multiple thin sections along high-pressured frozen multi-cellular sample. This proof of concept was conducted on *C.elegans*, a well-known model specimen in developmental and cell biology. Therefore, I strongly support the acceptance of this manuscript after resolving the minor issues, bellow:

1. The authors explain that cutting and transferring a vitrified block, using the lift out, is the most time-consuming part of the process. However, it is not mentioned how much time the current procedure saves and how slow would the conventional approach with lift-out would take?
2. While the authors reported on the number of specimen that are shown and were sectioned within the course of the study, it does not mentioned how reproducible the approach is (only 2 worms are mentioned, P.8)? Statistics on 2 specimen is on the lower side.
3. The left part of Fig. 4a is much darker then the right side. Wouldn't the contrast should be comparable?
4. P. 9, a field of view has 2D and not 1D
5. Almost no features are seen in the nucleus (almost not nucleosomes) neither of the nuclear pore complexes (Fig. 4b), how can this be explained?

6. Cryo-ET of lamellae from Ce worms were published before and therefore should be cited.

Author Rebuttal to Initial comments

We thank the reviewers and editor for their thorough comments and constructive feedback on our manuscript. Here, we will provide a point-by-point response to the reviewers' comments and highlight some of the changes we have made in response to both the editor's and reviewers' comments. Additionally, we added a detailed step-by-step protocol on the Serial Lift-Out method as well as the subtomogram average of the 11 protofilament microtubule structure (Figure 5, Figure Supplement 2, and p. 121. 301-309).

Reviewers' Comments:

Reviewer #1:

Remarks to the Author:

Cryo-ET on multicellular model organisms and organoids of all kind have great potential for discoveries and drug testing but this will only happen after many method development obstacles have been addressed properly. Plasma focused ion beam technology showed recently great potential for real structural cell biology and the sub nanometer resolution in lamellae of intact cells.

Cryo-lift-out procedure of lamellae of multicellular tissues as discussed in the study has been problematic because off the low success rate. Not many groups invested significant effort to overcome all these major problems accept a few pioneers in the field. The team of Jurgen Plitzko is clearly a great example of searching for novel methodology. This manuscript shows several milestones in method development to transform in a novel and creative way the boundaries of the lift-out for multi cellular samples.

Lift-Out was also initiated by the Plitzko / Baumeister team a few years ago and this study of Serial Lift-Out is one of the next steps. This team shows the ability to section in increments of one to four micrometers. This process has two steps that contribute to specimen loss: sectioning (~300-500 nm) and lamella milling. Serial Lift-Out sections and tomograms have been mapped back into context using other sources of volumetric data as illustrated.

This work is very original and significant for the entire field of cryo-EM. The work is very well presented and each paragraph is easy to understand for an insider. I regard the data of high quality and the figures are well presented. I support the validity, the reliability with the

robust conclusions without any hesitation.

Minor comments:

The statement in the text: “While the former yields samples that are easily FIB-milled, the sample thickness that can reliably be vitrified is limited to roughly 10 μm” contradicts and earlier study of this group (Mahamid et al., in Science - Visualizing the molecular sociology at the HeLa cell nuclear periphery) where it was stated: “clearly, the centre of the cell undergoes incompletely vitrification due to the heat transfer ...”. As long as we have no strong quantitative data of reproducible measurement of thickness it is better not to overstate.

We thank the reviewer for pointing out this common inaccuracy. We agree that without strong quantitative data, it is better not to overstate the possible vitrification depth by plunge freezing. We have changed the relevant sections accordingly (p. 2-3 l. 67-68).

Line 392: an error in the sentence: This like likely due to the reduction of lamella ...

Line 551: an error in the sentence: The the GIS was ...

We thank the reviewer for spotting these mistakes and have fixed the errors in the updated manuscript.

Reviewer #2:

Remarks to the Author:

This manuscript describes an important development that would allow the structural study of tissues by cryo-ET. The use of cryo-FIB as a tool for sample preparations of cells, is gaining momentum in the last several of years. It becomes a common practice to produce lamellae from individual vitrified cells. However, working with multi cellular specimen is not yet trivial and only few examples were previously demonstrated and published. To this end, and in order to develop a reproducible procedure, the authors developed an elegant strategy that facilitate cryo-ET study of multiple thin sections along high-pressured frozen multi-cellular sample. This proof of concept was conducted on *C.elegans*, a well-known model specimen in developmental and cell biology. Therefore, I strongly support the acceptance of this manuscript after resolving the minor issues, below:

1. The authors explain that cutting and transferring a vitrified block, using the lift out, is the most time-consuming part of the process. However, it is not mentioned how much time the current procedure saves and how slow would the conventional approach with lift-out would take?

The procedure of preparing lift-out sites and performing the actual lift-out stay relatively similar to previous approaches. The time that the current procedure saves is mainly in the site preparation, transfer and attachment step. The repetitive site preparation and lift-out steps are omitted. The lift-out procedure which was previously performed for multiple sites in a one-to-one ratio of sites/lamella took around one hour per site. This process is cut down to one to two hours for all lamellae prepared in Serial Lift-Out. The sectioning during 'lift-in' takes around 15-30 minutes per section for an experienced user, as can also be observed in the supplementary movies (Figure 2 – Movie Supplement 1, Figure 2 – Movie Supplement 2). Details are now also given in the supplementary protocol.

2. While the authors reported on the number of specimen that are shown and were sectioned within the course of the study, it does not mentioned how reproducible the approach is (only 2 worms are mentioned, P.8)? Statistics on 2 specimen is on the lower side.

We thank the reviewer for the comment on the reproducibility of the method. We do agree that the two worms mentioned are on the lower side regarding statistics. Therefore, we added another worm dataset to the manuscript (Figure 4 –Figure Supplement 1) from which we also performed the additional subtomogram analysis as discussed with the editor (Figure 5, Figure 5 – Figure Supplement 2). We would like to point out that Figure 2 – Figure Supplement 4 show the application of the Serial Lift-Out method to *D. melanogaster* egg chambers. Additionally, the approach has been reproduced at both the European Molecular Biology Laboratory, EMBL, Heidelberg, Germany (Mahamid lab) and the Max-Planck-Institute of Molecular Physiology, Dortmund, Germany (Raunser lab) during visits of some of the co-authors of the paper to these labs. Furthermore, researchers at the Rosalind Franklin Institute, Didcot, United Kingdom (Grange lab) have informed us that they were able to reproduce the method based solely on the paper without communication with us regarding the procedure. Thus, we are confident that the approach is indeed reproducible and hope that the attached protocol will make it accessible for researchers in the field to use the method in cryo-ET sample preparation.

3. The left part of Fig. 4a is much darker than the right side. Wouldn't the contrast should be comparable?

Indeed, the contrast should be comparable on the width axis of a lamella. To match the overview rotation with the reconstructed tomograms, we have rotated the lamella overview by 90°, thus the milling direction is the horizontal axis rather than the vertical axis, with the ion beam coming from the right. Therefore, the thickness gradient commonly observed in lamella milling due to beam divergence and reduced ablation rates resulting in a slightly wedge-shaped lamella is oriented horizontally rather than vertically, leading to the contrast gradient observed. We have added a sentence to the Figure legend of Figure 4 to indicate the milling direction and adjusted the contrast and brightness of Figure 4a.

4. P. 9, a field of view has 2D and not 1D

We thank the reviewer for noticing this error and have adjusted the manuscript accordingly. (p. 9 l. 230)

5. Almost no features are seen in the nucleus (almost not nucleosomes) neither of the nuclear pore complexes (Fig. 4b), how can this be explained?

We thank the reviewer for the comment. We agree entirely that we expect features of the nuclear pore complex and macromolecular complexes within the nucleus to be more pronounced. Likely, the lamella thickness is the limiting factor in the presented data as both lift-out and 'Waffle' milling preparation at similar thicknesses show the same featureless nucleus and a pore in the nuclear membrane with very little density that could be assigned to a nuclear pore complex. While we do not have experimental proof at this point, we suspect high molecular crowding in *C. elegans* L1 larvae leading to reduced contrast and, thus, a lack of features in those areas within the tomogram. To demonstrate that these features can indeed be visualized, we have added a slice through a tomogram of the nuclear periphery of a thinner lamella region to this reviewer response (Response to reviewers, Figure 1). We chose to leave the tomogram in Figure 4b unchanged for illustrative purposes as the shown tomograms all originate from a single lamella.

Figure 1: Tomogram of the nuclear periphery from double-sided attachment experiment 2. a. Tomographic slice of a denoised tomogram of the nuclear periphery of a *C. elegans* nucleus. Arrowheads indicate regions with features that are likely to be nucleosomes. The white rectangles represent the nuclear pore complexes that are shown in top view from the cytoplasmic side in **b** and **c**, and **c**, cytoplasmic top view of NPCs as indicated in **a**.

6. Cryo-ET of lamellae from *Ce* worms were published before and therefore should be cited. We apologize for the unintended omission and have added citations to the manuscript (p. 3 l. 74-75). While we extensively searched for previously published cryo-ET data on *C. elegans*, we would be grateful if the reviewer could point out other citations we might have unintentionally missed.

Decision Letter, first revision:

Dear Sven,

Thank you for submitting your revised manuscript "Serial Lift-Out – Sampling the Molecular Anatomy of Whole Organisms" (NMETH-A52401A). It has now been seen by the original referees and their comments are below. The reviewers find that the paper has improved in revision, and therefore we'll be happy in principle to publish it in Nature Methods, pending minor revisions to comply with our editorial and formatting guidelines.

TRANSPARENT PEER REVIEW

Nature Methods offers a transparent peer review option for new original research manuscripts submitted from 17th February 2021. We encourage increased transparency in peer review by publishing the reviewer comments, author rebuttal letters and editorial decision letters if the authors agree. Such peer review material is made available as a supplementary peer review file. Please state in the cover letter 'I wish to participate in transparent peer review' if you want to opt in, or 'I do not wish to participate in transparent peer review' if you don't. Failure to state your preference will result in delays in accepting your manuscript for publication.

ORCID

Sincerely,
Rita

Rita Strack, Ph.D.
Senior Editor
Nature Methods

Reviewer #2 (Remarks to the Author):

The authors have modified the manuscript in accordance to the reviewer comments. Therefore, I recommend to accept the manuscript as it is.

Final Decision Letter:

Subject: Decision on Nature Methods submission NMETH-A52401B

Message: 25th Oct 2023

Dear Sven,

I am pleased to inform you that your Article, "Serial Lift-Out – Sampling the Molecular Anatomy of Whole Organisms", has now been accepted for publication in Nature Methods. Your paper is tentatively scheduled for publication in our February print issue, and will be published online prior to that. The received and accepted dates will be April 28, 2023 and Oct 25, 2023. This note is intended to let you know what to expect from us over the next month or so, and to let you know where to address any further questions.

Over the next few weeks, your paper will be copyedited to ensure that it conforms to Nature Methods style. Once your paper is typeset, you will receive an email with a link to choose the appropriate publishing options for your paper and our Author Services team will be in touch regarding any additional information that may be required.

You will receive a link to your electronic proof via email with a request to make any corrections within 48 hours. If, when you receive your proof, you cannot meet this

deadline, please inform us at rjsproduction@springernature.com immediately.

Please note that *Nature Methods* is a Transformative Journal (TJ). Authors may publish their research with us through the traditional subscription access route or make their paper immediately open access through payment of an article-processing charge (APC). Authors will not be required to make a final decision about access to their article until it has been accepted. [Find out more about Transformative Journals](https://www.springernature.com/gp/open-research/transformative-journals)

Authors may need to take specific actions to achieve [compliance with funder and institutional open access mandates](https://www.springernature.com/gp/open-research/funding/policy-compliance-faqs). If your research is supported by a funder that requires immediate open access (e.g. according to [Plan S principles](https://www.springernature.com/gp/open-research/plan-s-compliance)) then you should select the gold OA route, and we will direct you to the compliant route where possible. For authors selecting the subscription publication route, the journal's standard licensing terms will need to be accepted, including [self-archiving policies](https://www.springernature.com/gp/open-research/policies/journal-policies). Those licensing terms will supersede any other terms that the author or any third party may assert apply to any version of the manuscript.

Your paper will now be copyedited to ensure that it conforms to Nature Methods style. Once proofs are generated, they will be sent to you electronically and you will be asked to send a corrected version within 24 hours. It is extremely important that you let us know now whether you will be difficult to contact over the next month. If this is the case, we ask that you send us the contact information (email, phone and fax) of someone who will be able to check the proofs and deal with any last-minute problems.

If, when you receive your proof, you cannot meet the deadline, please inform us at rjsproduction@springernature.com immediately.

Once your manuscript is typeset and you have completed the appropriate grant of rights, you will receive a link to your electronic proof via email with a request to make any corrections within 48 hours. If, when you receive your proof, you cannot meet this deadline, please inform us at rjsproduction@springernature.com immediately.

Once your paper has been scheduled for online publication, the Nature press office will be in touch to confirm the details.

Once your paper has been scheduled for online publication, the Nature press office will be in touch to confirm the details.

Content is published online weekly on Mondays and Thursdays, and the embargo is set at

16:00 London time (GMT)/11:00 am US Eastern time (EST) on the day of publication. If you need to know the exact publication date or when the news embargo will be lifted, please contact our press office after you have submitted your proof corrections. Now is the time to inform your Public Relations or Press Office about your paper, as they might be interested in promoting its publication. This will allow them time to prepare an accurate and satisfactory press release. Include your manuscript tracking number NMETH-A52401B and the name of the journal, which they will need when they contact our office.

About one week before your paper is published online, we shall be distributing a press release to news organizations worldwide, which may include details of your work. We are happy for your institution or funding agency to prepare its own press release, but it must mention the embargo date and Nature Methods. Our Press Office will contact you closer to the time of publication, but if you or your Press Office have any inquiries in the meantime, please contact press@nature.com.

Nature Portfolio journals [encourage authors to share their step-by-step experimental protocols](https://www.nature.com/nature-research/editorial-policies/reporting-standards#protocols) on a protocol sharing platform of their choice. Nature Portfolio 's Protocol Exchange is a free-to-use and open resource for protocols; protocols deposited in Protocol Exchange are citable and can be linked from the published article. More details can found at www.nature.com/protocolexchange/about.

Best regards,
Rita

Rita Strack, Ph.D.
Senior Editor
Nature Methods